



# Photoacoustic hygrometer for icing wind tunnel water content measurement: Design, analysis and intercomparison

Benjamin Lang[1,2,3], Wolfgang Breitfuss[4], Simon Schweighart[2], Philipp Breitegger[1], Hugo Pervier[5], Andreas Tramposch[2], Andreas Klug[3], Wolfgang Hassler[2], and Alexander Bergmann[1]

[1]Graz University of Technology, Institute of Electrical Measurement and Sensor Systems, Graz, Austria
[2]FH JOANNEUM GmbH, Institute of Aviation, Graz, Austria
[3]AVL List GmbH, Nanophysics & Sensor Technologies, Graz, Austria
[4]RTA Rail Tec Arsenal Fahrzeugversuchsanlage GmbH, Vienna, Austria
[5]Cranfield University, School of Aerospace, Transport and Manufacturing, Cranfield, United Kingdom

**Correspondence:** B. Lang (benjamin.lang@tugraz.at)

**Abstract.** This work describes the latest design, calibration and application of a near-infrared laser diode-based photoacoustic (PA) hygrometer, developed for total water content measurement in simulated atmospheric freezing precipitation and high ice water content conditions with relevance in fundamental icing research, as well as aviation testing and certification. The single-wavelength and single-pass PA absorption cell is calibrated for molar water vapor fractions with a two-pressure humidity generator integrated into the instrument. Laboratory calibration showed an estimated measurement accuracy better than $3.3\,\%$ in the water vapor mole fraction range of $510-12,360\,\mathrm{ppm}$ ($5\,\%$ from $250-21,200\,\mathrm{ppm}$) with a theoretical limit of detection ($3\sigma$) of $3.2\,\mathrm{ppm}$. The hygrometer is examined in combination with a basic isokinetic evaporator probe (IKP) and sampling system designed for icing wind tunnel application, for which a general description of total condensed water content (CWC) measurement and uncertainties are presented. Despite the current limitation of the IKP to a hydrometeor mass flux below $90\,\mathrm{g\,m^{-2}\,s^{-1}}$, a CWC measurement accuracy better than $20\,\%$ is achieved by the instrument above a CWC of $0.14\,\mathrm{g\,m^{-3}}$ in cold air ($-30\,^\circ\mathrm{C}$) with suitable background humidity measurement. Results of a comparison to the Cranfield University IKP instrument in freezing drizzle and rain show a CWC agreement of the two instruments within $20\,\%$, which demonstrates the potential of PA hygrometers for water content measurement in atmospheric icing conditions.

## 1 Introduction

Atmospheric water in the form of clouds and precipitation is of particular concern to aviation at temperatures below freezing, as supercooled liquid water and ice crystal environments present potentially hazardous conditions to aircraft, leading to airframe and air data probe icing (Vukits, 2002; Gent et al., 2000) or in-flight engine power loss (Mason et al., 2006).

Freezing precipitation containing supercooled large drops (SLDs), with drop diameters in excess of $50\,\mu\mathrm{m}$, as well as convective mixed-phase and glaciated clouds with high mass concentrations of ice crystals, i.e., ice water contents (IWCs) up to several grams per cubic meter, constitute two particular meteorological environments associated with severe icing events (Politovich, 1989; Bernstein et al., 2000; Cober et al., 2001b; Riley, 1998).





SLD icing environments of freezing drizzle (maximum drop diameters from $100\,\mu m$ to $500\,\mu m$) or freezing rain (max. diameters greater than $500\,\mu m$), as classified for the certification of large transport aircraft, are comprehensively characterized by envelopes of liquid water content (LWC), temperature, pressure altitude, drop size distributions and horizontal extent in Appendix O of the European Aviation Safety Agency Certification Specifications 25 (EASA CS-25, 2020) and the Code of
Federal Regulations Title 14 Part 25 (FAA CFR-25, 2019). Mixed-phase and ice crystal environments are likewise covered with a total condensed water content envelope by Appendix P and D of the two documents, respectively.

Replication of the full SLD, mixed-phase or high IWC condition envelopes in icing wind tunnels (IWTs) has been largely accomplished by organizations devoted to the experimental simulation of icing environments for the purpose of fundamental icing research and certification of aeronautical components, but is associated with a lack of appropriate instrumentation and is
still a work in progress for some conditions (Orchard et al., 2018; Van Zante et al., 2018; Bansmer et al., 2018; Breitfuss et al., 2019; Chalmers et al., 2019).

The accuracy and reliability of conventional water content instrumentation in the conditions encompassed by Appendix O and P/D is an issue frequently addressed for in-flight and IWT characterization (Strapp et al., 2003; Korolev et al., 2013; Orchard et al., 2019). Conventional instrumentation in this context refers to ice accretion blades or cylinders for LWC mea-
surement and evaporating (multi-element) hot-wire sensors used for simultaneous LWC and total condensed water content (CWC [1]; combined LWC and IWC) measurement. Both methods are either known or suspected to suffer from size and water content dependent inaccuracies in large drop or ice crystal icing environments due to uncertainties in collection efficiency and mass losses before accretion or evaporation (Cober et al., 2001a; Strapp et al., 2003; Emery et al., 2004; Isaac et al., 2006; Korolev et al., 2013; Steen et al., 2016).
This situation has led to the development of new benchmark isokinetic evaporator probe (IKP) instruments for CWC measurement (Davison et al., 2008; Strapp et al., 2016), regarded as closest to a first principles measurement and primarily designed for and deployed in the characterization of high IWC mixed-phase/glaciated conditions (e.g. Ratvasky et al., 2019). IKPs are used to extractively sample droplets and ice crystals in the icing environment with a forward facing, isokinetically operated inlet. After sampling, hydrometeors are evaporated to measure the combined condensed and ambient air water content with a
suitable hygrometer. Ambient air background water vapor (BWV) is measured separately and subtracted from the total water content (TWC) to derive the condensed water content. Measurement of the BWV concentration is usually accomplished via a second, backward facing inlet connected to another hygrometer. Due to the isokinetic sampling, losses of droplets or particles by re-entrainment into the flow after entering a sufficiently long inlet are improbable. Hence, IKP particle size distribution dependence is in theory only governed by the aspiration efficiency of the inlet.
Collectively, only few such reference instruments for CWC measurement in icing conditions similar to Appendix O and P/D currently exist. This lack of instrumentation has motivated the development of the hygrometer and sampling system described in this work.

Hygrometers in devices specifically designed for IWT operation typically apply commercially available optical absorption spectroscopy based non-dispersive infrared (NDIR) gas analyzers (e.g. Strapp et al., 2016; Bansmer et al., 2018, sec. 4.3). The

---

[1]Often abbreviated as *TWC*. To provide a clear distinction to total water content, we adhere to the nomenclature and reasoning given by Dorsi et al. (2014).





upper end of water contents that have to be within the range of suitable hygrometers is given by the combined background and condensed water content in the measurement environment. The former is approximately limited to fully saturated air at a static air temperature (SAT) of $0\,°C$ and the latter may be taken as an upper bound of $10\,g\,m^{-3}$ to the peak CWC of $9\,g\,m^{-3}$ in high IWC conditions (EASA CS-25, 2020). This may add up to molar water vapor fractions of $18,500\,ppm$ at standard pressure

$(1000\,hPa)$. Accuracy requirements are primarily determined by high BWV concentrations that have to be subtracted from high total water concentrations at low CWC and high ambient temperatures (Davison et al., 2016). The necessary hygrometer limit of detection highly depends on the specific measurement conditions but may be estimated from the fact that detection of a CWC of $0.05\,g\,m^{-3}$ in dry air at standard pressure requires an accuracy and limit of detection better than $48\,ppm$.

With the measurement system described in detail by Szakáll et al. (2001), Tátrai et al. (2015) have first demonstrated

the suitable accuracy of photoacoustic (PA) hygrometers in and beyond the above measurement range. Compared to NDIR sensors, photoacoustic spectroscopy offers the potential of achieving higher signal to noise ratios (SNRs) with equal response time, while providing high selectivity and high robustness, due to the possibility of optical single-pass arrangements and an instrument response that is invariant to the total absorption path length (Hodgkinson and Tatam, 2013).

In this work we describe the latest design, preliminary calibration and basic properties of a new PA hygrometer and two-

pressure humidity generator, developed with the goal of providing the total water measurement and calibration ranges typical for simulated atmospheric icing conditions applied in aviation testing and certification. The hygrometer is examined in combination with a basic IKP and sampling system, designed for IWT application in Appendix O conditions, for which a description of CWC measurement and associated uncertainties are presented. Finally, results of water content measurements in freezing drizzle and rain conditions in a closed circuit IWT, calibrated according to SAE Aerospace Recommended Practices (SAE

ARP-5905, 2015), are presented and compared to measurements with a reference IKP and a hot-wire instrument.

## 2 Instrument design

A schematic overview of the entire instrument is shown in Fig. 1(a). The system consists of a sampling probe positioned inside the icing wind tunnel and a measurement and sampling unit integrated into a 19-inch rack, positioned outside the tunnel and connected by $7\,m$ long heated and thermally insulated PTFE tubing, temperature-controlled to the measurement temperature

of $35\,°C$ to prohibit condensation. The probe is a total water (TW) sampling probe operated isoaxially and near isokinetic conditions, also featuring a second inlet port intended for BWV measurement. Hydrometeors entering the forward facing TW inlet are evaporated inside the probe, enriching simultaneously sampled ambient air by the evaporated condensed water.

The sampling system is designed to provide five main operating modes:

1. TW measurement (Path 1 in Fig. 1(a)),

2. BWV measurement (Path 2),

3. Zeroing (PA background signal measurement; Path 3),

4. Calibration (Path 4),





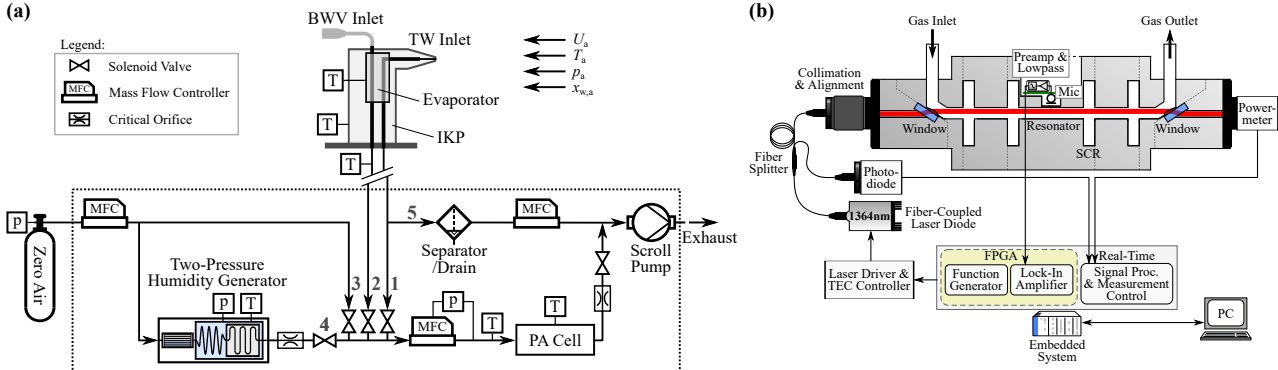

**Figure 1. (a)** Schematic of the instrument showing the isokinetic evaporator probe (IKP) and the measurement (PA cell), sampling and calibration system (two-pressure humidity generator and zero air). Locations of temperature and pressure control are indicated by (T) and (p), respectively. Indicated numbers enumerate the individual flow paths. The rearward facing BWV probe indicates the extension of the BWV inlet port not applied in this work. **(b)** Schematic of the photoacoustic cell together with the optical and the electronic setup, showing the control, data acquisition and signal processing performed on the real-time embedded system.

5. Inlet purging (Path 3 combined with Path 1 or 2).

For TW and BWV measurement, air sampled through the respective inlet is continuously pumped to the measurement unit, where the PA hygrometer (PA cell) is used to measure the water vapor mole fraction in parts of the TW or the full BWV inlet air flow. Currently, only a single hygrometer has been implemented, and humidity measurement may only be alternated between

5   TW and BWV measurement.

During isokinetic TW sampling the majority of the flow is bypassed the hygrometer (path 5 in Fig. 1(a)) to the scroll pump (Edwards, nXDS10iC). The hygrometer is supplied by a constant standard volumetric flow rate of 0.75(4) standard $1\,\mathrm{min}^{-1}$ (slpm; reference conditions: $273.15\,\mathrm{K}$ and $1013.25\,\mathrm{hPa}$), set by a pressure controller (Vögtlin Instruments, GSP-B9SA-BF26) upstream the cell and a critical orifice of $350\,\mu\mathrm{m}$ nominal diameter downstream the cell. A calibrated mass flow controller

10  (MFC; Vögtlin Instruments, GSC-C9SA-FF12) is used to control the bypass flow rate, and a calibrated flow meter included in the pressure controller measures the actual hygrometer flow rate. Isokinetic sampling at the TW inlet is set by adjusting the MFC flow rate to a combined flow rate matching isokinetic conditions, which are calculated using IWT test section operating and TW inlet geometry parameters (cf. Section 4).

The instrument features a two-pressure humidity generator also integrated into the rack, which in combination with zero air

15  is used for calibration and zeroing of the hygrometer. Control of flow, temperature and pressure together with signal processing and data logging for the sampling system and humidity generator is performed with a dedicated embedded system (National Instruments, NI cRIO 9063).

In the following subsections, the major components of the instrument are described in further detail.





## 2.1 Photoacoustic hygrometer

The hygrometer is a custom built single-cell photoacoustic absorption spectrometer, providing a signal proportional to the water vapor number concentration in the total water or background water air stream. Figure 1(b) presents a schematic of the PA cell together with the optic configuration and electronic setup.

A fiber-coupled distributed feedback laser diode (NEL, NLK1E5GAAA) is intensity modulated at approximately $4584\,\mathrm{Hz}$ (at $35\,°\mathrm{C}$) to excite the fundamental acoustic resonance mode of the PA cell when water vapor is present. The diode is temperature-controlled to the peak of a ro-vibrational water vapor transition at $1364.68\,\mathrm{nm}$ ($7327.7\,\mathrm{cm^{-1}}$; 296 K), which was chosen based on HITRAN simulations (Gordon et al., 2017) as it exhibits minimal line shift with pressure, high absorption cross section and low interference from other anticipated atmospheric constituents. Intensity modulation is performed by square wave modulating

the applied laser current at the resonance frequency from the maximum permissible laser diode current down to just below the lasing threshold with a benchtop laser driver (Thorlabs, ITC4001), maintaining an average optical power of $9.9(1)\,\mathrm{mW}$. The laser beam is collimated to a diameter of $2\,\mathrm{mm}$ and directed through the resonator via two N-BK7 Brewster windows angled at $56.4\,°$. A thermal powermeter (Thorlabs, PM16-401) is used to measure average optical power when the cell is flushed with zero air during PA background signal measurements. Monitoring of the laser power during measurements is accomplished

by a fiber splitter with a $99:1$ split ratio (Thorlabs, TW1300R1A1) in combination with a temperature-controlled InAsSb photodetector (Thorlabs, PDA10PT-EC). However, the high wavelength and output power stability of the laser diode allows stable operation over the duration of typical measurement series, thus no wavelength locking on the absorption line or power correction is applied on measured signals in between calibration cycles.

    Measurement air is pumped through the stainless steel PA cell via milled $6\,\mathrm{mm}$ inner diameter (ID) cylindrical ducts. At

the center of the modularly designed cell a $34\,\mathrm{mm}$ long cylindrical resonator is formed by a termination on either side with two acoustically short concentric resonators (Selamet and Radavich, 1997). Short concentric resonators are used instead of larger expansion chambers (buffer volumes) to decrease gas exchange and measurement response time. The diameters and distances in between the small volume acoustic band-stop filters are tuned to maximize resonator quality factor ($Q{=}17$), while minimizing transmission of external noise into the cell. At the center of the resonator and the location of the antinode of the

fundamental longitudinal resonance mode, an electret condenser microphone (Knowles, EK-23028) is connected in a small volume gas- and noise-tight enclosure to measure the PA pressure signal.

    The PA cell is operated at constant temperature, pressure and flow to maintain a microphone sensitivity and resonance frequency independent of ambient and IWT conditions.

    The temperature of the thermally insulated PA cell is controlled to $35.0(3)\,°\mathrm{C}$ by two integrated heating cartridges to sta-

bilize resonance frequency and microphone sensitivity[2]. An additional resistance temperature detector (RTD), installed in the sampling gas stream approximately $100\,\mathrm{mm}$ upstream of the cell, is used to control the gas temperature to $35.0(3)\,°\mathrm{C}$ inside the PA cell by controlling the heating of the upstream tubing in the measurement unit. This temperature also sets the theoretical

---

[2]The number in parenthesis gives the half-width of the rectangular confidence interval in terms of the last digit.



upper water vapor mole fraction measurement limit of $58,600\,\mathrm{ppm}$ before condensation of water vapor in the sampling lines and the PA cell occurs.

Although the sampling system and the IKP are designed to operate around standard pressure, the PA cell pressure may be set with the pressure controller upstream of the hygrometer within the limits given by the pressure loss of the upstream flow

elements down to $100\,\mathrm{hPa}$. For IWT measurement, cell pressure is set to $800(8)\,\mathrm{hPa}$, close to the pressure of optimal signal-to-noise ratio (SNR). Optimum measurement pressure is primarily defined by the valve position of the pressure controller, due to flow noise generated at the valve. To further decrease signal noise, the PA cell is vibrationally decoupled from the scroll pump mounted in the rack by a vibration absorbing mount and short sections of PTFE-tubing at the gas in- and outlet of the cell.

Laser current control, signal processing and data logging of microphone and power monitoring signals is carried out with a second dedicated embedded system (National Instruments, NI cRIO 9031), a real-time processor combined with a reconfigurable field programmable gate array (FPGA). The laser current modulation signal is generated by a function generator implemented on the FPGA Data acquisition of the microphone signal after analog amplification with a transimpedance amplifier (10-fold gain), together with the photodetector signal is carried out with a $24\,\mathrm{bit}$ ADC (National Instruments, NI 9234) at

a sampling rate of $52.1\,\mathrm{kHz}$. A digital dual-phase lock-in amplifier implemented on the FPGA is used to determine in-phase and quadrature components of the microphone signal at the frequency of modulation. The lock-in signal amplitude (referred to as PA signal), used to derive the water vapor mole fraction, is calculated and logged on the real-time operating system with a $10\,\mathrm{Hz}$ rate after phase-correct background signal correction (cf. Appendix A).

Despite operation at controlled measurement conditions, the hygrometer sensitivity is a function of the measured water con-

tent due to several reasons. Increasing water contents cause decreasing irradiance along the absorption path (Beer-Lambert law) and therefore reduce sensitivity. In addition, the electret microphone sensitivity is a function of humidity (specified $0.02\,\mathrm{dB}\,\%\mathrm{RH}^{-1}$; Langridge et al., 2013). Furthermore, speed of sound and therefore also resonator resonance frequency is a function of humidity (Zuckerwar, 2002). Shifts in resonance frequency may reduce effective resonator amplification and sensitivity according to the approximately Lorentzian resonator frequency response, if the frequency of modulation is not shifted

accordingly (Szakáll et al., 2009). Finally, photoacoustic conversion efficiency (i.e., conversion of absorbed laser radiation to a detectable pressure signal) for water vapor in air is concentration dependent and over the range of typical atmospheric concentrations and pressures varies by a factor of five (Lang et al., 2020).

All above effects are to a great extent accounted for by calibrating the hygrometer over the range of expected water vapor concentrations and by applying a suitable nonlinear calibration function, which is described in greater detail in Lang et al.

(2020). The PA signal reduction associated with resonance frequency humidity dependence ($0.5\,\%$ for the $14\,\mathrm{Hz}$ shift from $0\,\mathrm{ppm}$ to $20\,000\,\mathrm{ppm}$) is taken into account by maintaining the laser modulation frequency at the dry air resonance frequency ($4584\,\mathrm{Hz}$ at $35\,^{\circ}\mathrm{C}$) for calibration and measurements. This method results in maximum amplification and PA signal at low concentrations. The approximately quadratic sensitivity loss for higher concentrations is considered in the second order term of the calibration function.





## 2.2 Calibration unit

Determination of the water vapor concentration from the hygrometer signal requires background signal correction (zeroing) and calibration with known concentrations of water vapor. The system is calibrated by generating and providing a stable flow of humidified air with known molar fractions of water vapor to the inlet of the hygrometer (e.g., Dorsi et al., 2014; Tátrai et al.,

2015). This approach is preferred to the method of introducing a continuous stream of liquid water or ice into the TW inlet and calibrating for CWC (e.g., Strapp et al., 2016), as calibration may be performed during IWT operation without removing the sampling probe. With the goal of performing calibration over a major part of the necessary water content range within a short time, a compact custom-made two-pressure humidity generator (HG) has been integrated into the instrument. Two-pressure humidity generation offers the benefit of enabling rapid and accurate setting of a wide range of humidity levels in a saturation

chamber at a convenient and constant temperature by varying the pressure and thus the molar water vapor fraction (Wernecke and Wernecke, 2013).

Zeroing of the instrument is performed by acquiring a PA background signal after continuously flushing the PA cell with zero air from an external gas cylinder (Messer, scientific grade synthetic air; residual water volume fraction below $2\,\mathrm{ppmv}$) until a stable reading is attained (approx. $20\,\mathrm{min}$).

For calibration zero air is initially humidified in a pre-saturation stage, a porous ceramics with honeycomb structure (IBIDEN Ceram) in a room temperature water bath, to a dew point well above the main saturation chamber dew point. The humidified air is subsequently passed through a lower temperature and pressure-controlled $1\,\mathrm{m}$ long coiled tube heat exchanger and the main saturator, where the air is saturated with respect to the local temperature and pressure.

The saturator is a $(6\times25\times600)\,\mathrm{mm}$ (width$\times$height$\times$length) channel milled into a stainless steel block, hermetically sealed

and partially filled with distilled water. Both, heat exchanger and saturator are placed in a stirred and thermally insulated water bath, temperature-controlled by thermoelectric coolers within the range of $1\,^\circ\mathrm{C}$ and ambient temperature. Saturator air pressure is controlled within the range of $1000\,\mathrm{hPa}$ to $8000\,\mathrm{hPa}$ with an MFC (Vögtlin Instruments, GSC-C9SA-FF12) upstream of the HG. By increasing the saturator pressure to its maximum value, the $1000\,\mathrm{hPa}$ water vapor saturation fraction may be reduced by a ratio of $1:7.8$.

Bath temperature and saturator air pressure are measured with a high precision four-wire Pt100 (Omega Engineering, P-M-1/10-1/8-6-0-PS-3) combined with a calibrated 24-Bit ADC (National Instruments, NI 9217) and a calibrated pressure transducer (KELLER AG, PAA 33X), traceable to NIST and Swiss national standards, respectively. Associated measurement uncertainties are given in Table C1. The molar water vapor saturation fraction, which remains constant during expansion to the lower pressure level of the hygrometer, is calculated from the measured saturation temperature and pressure according to

Wagner and Pruss (1993) and Greenspan (1976).

In the described configuration the operational range of the HG extends from $845\,\mathrm{ppm}$ to approximately $22,000\,\mathrm{ppm}$ (maximum saturator temperature of $19\,^\circ\mathrm{C}$). Two saturator temperature setpoints are used for calibration, covering the full humidity range by varying the saturator pressure. The settling time to a stationary hygrometer signal after changes in the HG settings is





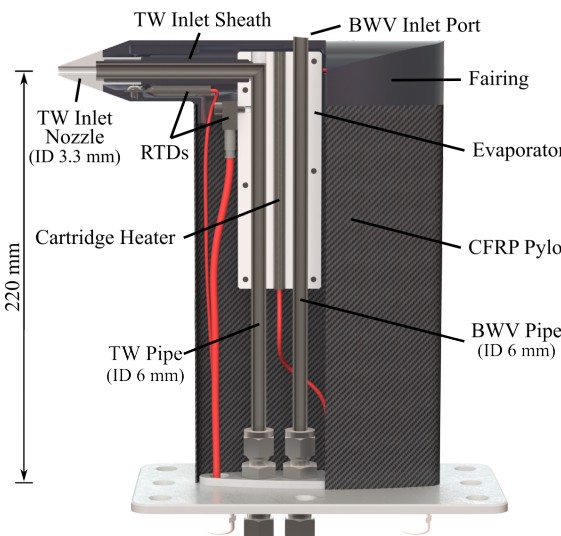

**Figure 2.** Schematic of the isokinetic evaporator probe assembly showing a partial cut through the main components: TW and BWV inlet lines with TW inlet nozzle, the carbon fiber reinforced polymer (CFRP) pylon, aluminum fairing and evaporator. Fairing cartridge heaters extending alongside the TW inlet sheath are not indicated.

below 7 min. This figure is mainly determined by the relatively low signal noise of the PA hygrometer compared to the slow water vapor adsorption-desorption processes at the piping and cell walls.

An independent calibration of the humidity generator is still pending, which in particular is necessary to verify full saturation at high loads (high saturator temperature). To asses the HG and thus also hygrometer accuracy, the uncertainty in the generated

humidity is calculated from first principles, i.e., the measured saturator temperature and pressure and the associated uncertainties, according to Meyer et al. (2008). The resulting uncertainty (95 %) is below $\pm 2.1\,\%$ over the entire range of humidities provided by the HG and is dominated by the saturator temperature measurement uncertainty (cf. Table C1).

### 2.3 Isokinetic evaporator probe

The inlet system has been designed around the three requirements of enabling reasonably representative isokinetic TW sam-

pling while providing the necessary heating power for hydrometeor evaporation and maintaining the probe free from ice accretion at high water contents. The probe inlets are housed in an airfoil-shaped $(32 \times 132)\,\mathrm{mm}$ (width×length) carbon fiber reinforced polymer (CFRP) pylon capped by an additively manufactured aluminum fairing, with the TW centerline extending 220 mm perpendicular to the free-stream flow from a $(100 \times 195)\,\mathrm{mm}$ base flange. A CAD drawing of the IKP is shown in Fig. 2.

The fairing is controlled to a TW inlet nozzle temperature of approximately $50\,^{\circ}\mathrm{C}$ by maintaining constant $80\,^{\circ}\mathrm{C}$ at the RTD (Pt100) inside the fairing front tip. To this end, integrated cartridge heaters in the aluminum enclosure provide a maximum combined heating power of $390\,\mathrm{W}$.





### 2.3.1 Total water inlet

TW is sampled through a screw-on aluminum nozzle with a sharp leading edge and a tapering half angle of $20°$. For the measurements presented, a nozzle with an inlet inner diameter of $3.30(15)\,\text{mm}$, measured with a standard caliper, was used. The particular choice of the comparatively small inlet diameter is based on the maximum continuous flow rate attainable

with the low-noise vacuum pump in use, which in combination with the TW inlet area determines the maximum wind tunnel airspeed for which isokinetic sampling may be maintained. The ID of $3.3\,\text{mm}$ corresponds to a maximum airspeed slightly above the main targeted wind tunnel airspeed of $60\,\text{m s}^{-1}$. The stated nozzle inner diameter uncertainty is attributed to the measurement method and measurable inlet deformations caused by the machining process.

As the TW inlet is considered a thick-walled inlet with an aspiration efficiency expected to deviate from an ideal sampling

behavior (Belyaev and Levin, 1974), the collection efficiency of the inlet was determined from combined computational fluid dynamics (CFD) and Lagrangian particle tracking simulations. Definitions of aspiration and collection efficiency, as well as the particle Stokes number $St_\text{p}$ used in the evaluation, are given in Appendix B. Simulations were carried out in COMSOL Multiphysics software with a workflow similar to the one described by Krämer and Afchine (2004) and showed good agreement with simulations carried out in ANSYS CFX for the same probe with an inlet diameter of $4.6\,\text{mm}$. However, instead of

determining the limiting freestream area $A_\text{lim}$ comprising all particle trajectories entering the inlet, collection efficiencies $E(d_\text{p})$ for each droplet diameter $d_\text{p}$ considered were calculated from the ratio of the number $N_\text{s}$ of droplets sampled to the number $N_\text{inlet}$ of droplets passing through the probe TW inlet equivalent area $A_\text{inlet}$ in freestream (cf. Appendix B):

$$E(d_\text{p}) = \frac{A_\text{lim}}{A_\text{inlet}} \approx \frac{N_\text{s}}{N_\text{inlet}} \ . \tag{1}$$

Figure 3 shows the determined collection efficiencies for two IWT freestream airspeeds $U_\text{a}$ and different isokinetic factors

$\text{IKF} = \overline{U}_\text{s}/U_\text{a}$, i.e., velocity ratios of mean inlet sampling velocity $\overline{U}_\text{s}$ to freestream airspeed. Low collection efficiencies at Stokes numbers around one are the result of the thick-walled inlet design (Rader and Marple, 1988). At the conditions of the measurements presented herein ($U_\text{a} = 60\,\text{m s}^{-1}$ and $\text{IKF} \approx 1$), the simulated collection efficiency reaches a minimum of $88\,\%$ for particles of $3\,\mu\text{m}$ diameter and is practically independent of the IKF in the range of $0.95$ to $1.05$ for diameters above $10\,\mu\text{m}$ ($St_\text{p} \approx 7$). For $St_\text{p} \ll 1$, the collection efficiency tends towards the value of the isokinetic factor. Consequences of the

non-representative sampling on cloud CWC measurement depend on the individual particle size distribution and are discussed in further detail in Section 4.3.

Hydrometeors aspirated through the TW inlet are transported down a $6\,\text{mm}$ inner diameter stainless steel tubing to the evaporator, a $(125{\times}44{\times}16)\,\text{mm}$ aluminum block controlled to $180\,°\text{C}$ by a $400\,\text{W}$ cartridge heater. An aluminum sheath connects the evaporator and the nozzle and ensures additional heat transfer from the evaporator to the inlet. A sharp $90°$ bend

of the tubing approximately $100\,\text{mm}$ downstream the inlet forms an impactor, where larger droplets and particles are impacted on the heated wall to increase heat transfer and promote droplet or particle break-up. At the bend the piping is enclosed and in good thermal contact with the evaporator.

For the airspeed of $60\,\text{m s}^{-1}$ and the conditions of the measurements presented, calculated particle stopping distances $S_\text{p}$ (cf. Appendix B) predict impaction at the bend for particles with diameters larger than approximately $15\,\mu\text{m}$. This is in close



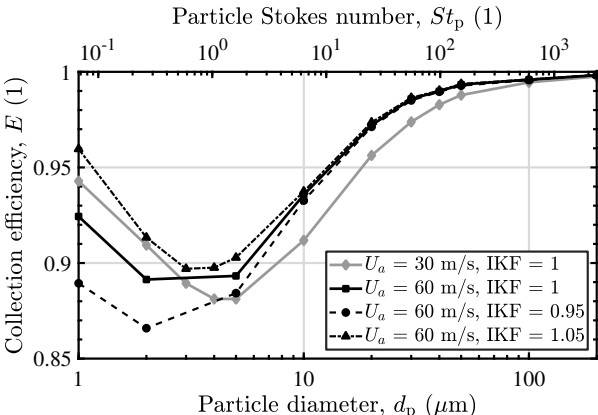

**Figure 3.** Isoaxial TW inlet collection efficiency as a function of particle diameter determined from combined CFD and Lagrangian particle tracking simulations at different freestream airspeeds $U_a$ and isokinetic factors (IKFs), assuming an ambient air temperature and pressure of $-5\,°C$ and $1013.25\,hPa$, respectively. Particle Stokes numbers given in the upper x-axis are only valid for $60\,m\,s^{-1}$ data. The lines between the evaluation points are used to guide the eye.

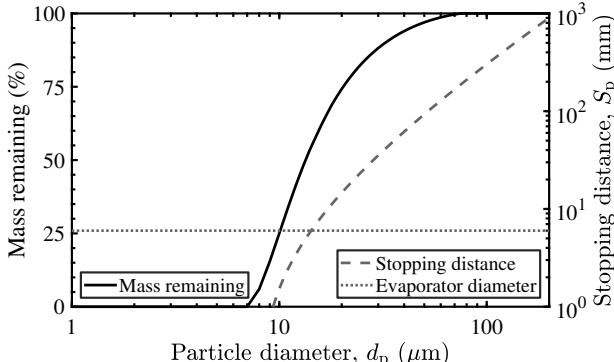

**Figure 4.** Calculated droplet mass remaining after traversing the probe TW pipe following the evaporator bend and stopping distance as a function of the initial droplet diameter, assuming an ambient air temperature, pressure and freestream airspeed of $-5\,°C$, $1013.25\,hPa$ and $60\,m\,s^{-1}$, respectively. The indicated evaporator diameter marks the stopping distance equal to the inlet pipe diameter of $6\,mm$.

agreement with the CFD and Lagrangian particle tracking calculations. The calculated stopping distance in dependence of the particle diameter is shown in Fig. 4 together with the stopping distance equal to the evaporator pipe diameter (dotted line).

Also shown is a theoretical calculation of the evaporative mass loss of supercooled spherical droplets when passing the heated probe pipe section following the $90\,°$ bend. Droplet evaporation was calculated with the two-parameter model (droplet mass and temperature) summarized by Davis et al. (2007), which includes diffusion of water vapor from the droplet to the humid inlet air, associated latent heat losses and conductive heating of the droplet by the heated inlet air. For the computations



a minimum (centerline) air temperature of $50\,°C$ was assumed, which was determined from the CFD and heat transfer analysis. Inlet ambient air was assumed fully saturated at $-5\,°C$, with an additional worst case evaporated cloud CWC of $10\,\mathrm{g\,m^{-3}}$.

Droplets with diameters above $15\,\mu m$ impact the $180\,°C$ evaporator walls and are assumed to evaporate due to the increased heat transfer or break up into smaller, more easily evaporated droplets. Minimum residence times of $1\,s$ in the attached $7\,m$ long tubes heated to $35\,°C$ are considered sufficient to achieve full evaporation of smaller droplets. However, observable TW signal oscillations for inlet condensed water mass flow rates above $0.8\,\mathrm{mg/s}$ (hydrometeor mass flux of approx. $90\,\mathrm{g\,m^{-2}\,s^{-1}}$) suggest temporary accumulation of water or ice in the small diameter nozzle or at the evaporator and are the reason for further investigation into the process of droplet and particle evaporation for the chosen inlet diameter and evaporator geometry.

### 2.3.2 Background water vapor inlet

The BWV inlet port is used for sampling of ambient air with the PA cell mass flow rate of $0.75\,\mathrm{slpm}$ and may be extended by a rearward facing probe with a $16\,\mathrm{mm}$ ID connected to a $4\,\mathrm{mm}$ ID tubing. The connection between the rearward facing probe and the port has been thermally insulated to reduce heating of the inlet, as the port pipe is in direct contact with the evaporator. For the measurements presented, only a single hygrometer used for TW measurement was available, thus the IKP was used without the rearward facing probe and BWV was estimated from IWT humidity sensors. The method of BWV estimation is described in further detail in Section 5.

## 3 Hygrometer characterization and calibration

### 3.1 Noise and limit of detection

To quantify measurement noise, expectable system drift and the limit of detection (LOD) of the hygrometer an Allan deviation analysis (Werle et al., 1993) was performed on a background measurement with zero air, acquired at $10\,\mathrm{Hz}$ with an integration time of $0.1\,s$. Figure 5 shows the Allan deviation $\sigma_A$, i.e., an estimate for the standard deviation of the mean of the background signal, in dependence of the averaging or integration time $\tau$.

The system exhibits a $1/\sqrt{\tau}$ decrease in noise, typical for white noise averaging, up to a maximum useful averaging time of $150\,s$, where drift starts to deteriorate system performance. The effectiveness of increasing integration time is limited by a slow drift of the measurement gas temperature. For half the maximum useful averaging time an LOD ($3\sigma_A$), calculated from the calibration curve (see Section 3.2), of $3.2\,\mathrm{ppm}$ water vapor mole fraction or $2.0\,\mathrm{mg\,kg^{-1}}$ in terms of humid air mass mixing ratio at standard temperature and pressure (STP; $273.15\,\mathrm{K}$ and $1000\,\mathrm{hPa}$) can be achieved. More practical averaging times of $1\,s$ and $10\,s$ result in noise equivalent concentrations of $23\,\mathrm{ppm}$ and $7\,\mathrm{ppm}$, respectively.

As the $1\,s$ averaging time precision — equivalent to $14\,\mathrm{mg\,kg^{-1}}$ mass mixing ratio at STP — is sufficient for IWT water content measurement and results in favorable response time, this lock-in integration time is applied in calibration and water content measurements.

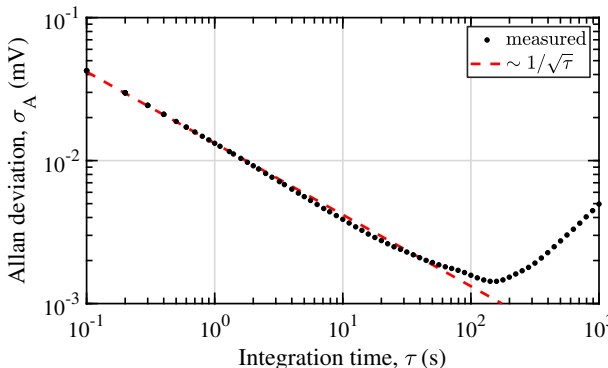

**Figure 5.** Allan deviation $\sigma_\mathrm{A}$ calculated from the measured signal amplitude of a one hour background measurement with zero air as a function of the lock-in integration time $\tau$. The dotted line indicates a $1/\sqrt{\tau}$ decrease in noise, typical for white noise averaging.

## 3.2 Hygrometer calibration

The hygrometer is calibrated at constant PA cell temperature and pressure ($800\,\mathrm{hPa}$, $35\,^\circ\mathrm{C}$) with the built-in two-pressure HG. To quantify measurement uncertainties at dew points lower than provided by the HG, a gas diluter (Breitegger and Bergmann, 2018) was used for an initial laboratory calibration. Using the gas diluter, humidified air provided by the humidity generator

was further diluted with zero air, down to a minimum water vapor mole fraction of $124\,\mathrm{ppm}$. Background corrected calibration data recorded at concentrations in the range of $124\,\mathrm{ppm}$ to $22,150\,\mathrm{ppm}$ and the inverse calibration curve used to determine the water vapor mole fraction during water content measurement is shown in Fig. 6. Signal amplitude noise of the hygrometer during calibration is typically below water vapor mole fractions of $10\,\mathrm{ppm}$ or $0.7\,\%$ (the higher value in absolute terms applies). The former value, applicable at low concentrations, is in the order of the background signal noise ($1\,\sigma$) determined by the Allan

deviation analysis for the integration time of $1\,\mathrm{s}$.

For the determination of the water vapor mole fraction during water content measurement, the calibration data is approximated by the inverse of the theoretically motivated nonlinear 5-parameter calibration function given by Lang et al. (2020), which accounts for the humidity-dependent hygrometer sensitivity. As opposed to higher-order polynomials, which are necessary to reproduce the nonlinear functional relationship, this calibration function adds the benefit of a well-defined behavior for

inter- and extrapolation when faced with a reduced number of calibration points. The parameters $\boldsymbol{b}$ of the calibration curve are determined with the weighted nonlinear least-squares method, minimizing

$$\chi^2 = \sum_{i=1}^{N} w_{x,i} \left[ x_{\mathrm{w},i} - f^{-1}(S_i, \boldsymbol{b}) \right]^2 \tag{2}$$

over the $N$ calibration measurements, where $f^{-1}(S_i, \boldsymbol{b})$ is the inverse calibration function evaluated at the measured PA signal amplitude $S_i$ and for the parameter set $\boldsymbol{b}$. The inverse of the calibration function has been used in order to include the uncertainty

of the calibration water vapor mole fraction $u(x_{\mathrm{w},i})$ in the determination of the parameters and parameter confidence intervals.

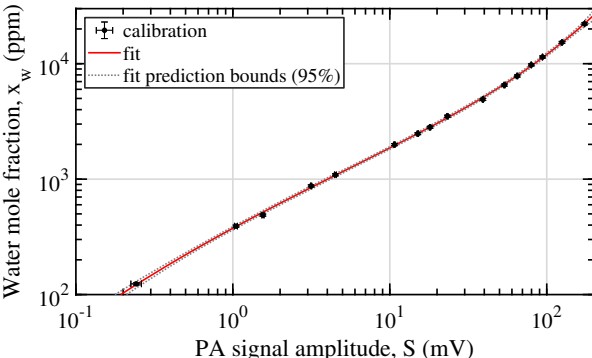

**Figure 6.** Laboratory calibration data of the PA hygrometer operated at $35\,°C$, $800\,hPa$ and an integration time of $1\,s$. Calibration humidities were set with the internal humidity generator and in combination with the gas diluter. The fit indicates the best-fit calibration curve with the parameters obtained by the weighted nonlinear least-squares method. Error bars of the measurements indicate the $95\,\%$ uncertainty of the humidity generation and standard deviation of the lock-in signal for the y- and x-axes, respectively.

To this end, each calibration point $i$ is weighted by $w_{x,i} = 1/\sigma_{x,i}^2 = 1/u^2(x_{w,i})$, i.e., according to the combined uncertainty in the humidity provided by the humidity generator and gas diluter. The uncertainty in the mean of the measured PA signal amplitude is negligible in comparison to the uncertainty in the mole fraction and therefore is disregarded in the least-squares fit. Residuals, i.e., the differences between calibration data and calibration curve, are typically below $3\,\%$. This remaining
variability is largely explained by the error in the generated humidity and changes in microphone sensitivity from temperature oscillations of the PA cell.

### 3.3    Estimation of hygrometer measurement uncertainty

The measurement uncertainty of the PA hygrometer is the result of uncertainties originating from the calibration and from noise during measurement. Calibration uncertainty itself includes uncertainties from humidity generation and from the approximation
by the calibration function. These uncertainties have been jointly estimated from the parameter uncertainties obtained with the nonlinear least-squares method. Instrument signal noise ($1\sigma$) is taken equivalent to the calibration noise ($10\,ppm$ or $0.7\,\%$, whichever is higher). Details to the determination of the combined hygrometer uncertainty are given in Appendix D of this work.

The calculated relative measurement uncertainty of the hygrometer ($95\,\%$ coverage) as a function of the measured water va-
por mole fraction is shown in Fig. 7. Measurement uncertainty can be seen to increase rapidly for mole fractions below $200\,ppm$ and above $23,000\,ppm$, due to the lack of calibration points at lower and higher water vapor concentrations. Nevertheless, in the range of expected condensed water contents and background humidities encountered during typical IWT evaluation, the hygrometer exhibits an accuracy better than $2.5\,\%$ to $3.3\,\%$. This target water content range is defined by the lower limit of cloud-free, but fully saturated air (with respect to supercooled liquid) at $-30\,°C$ and the upper limit of $5\,g\,m^{-3}$ in fully satu-





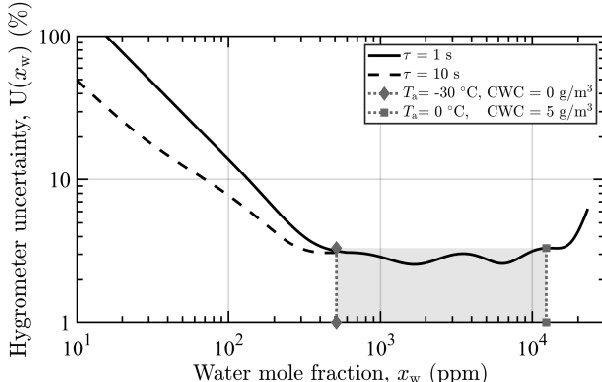

**Figure 7.** Relative measurement uncertainty (95 % coverage) of the photoacoustic hygrometer operated at $35\,°\mathrm{C}$, $800\,\mathrm{hPa}$ and with integration times $\tau$ of $1\,\mathrm{s}$ and $10\,\mathrm{s}$. The gray area bounded by dotted vertical lines marks the target range of background and total water contents, defined by the lower limit CWC of $0\,\mathrm{g\,m^{-3}}$ at $-30\,°\mathrm{C}$ ($512\,\mathrm{ppm}$) and the upper limit of $5\,\mathrm{g\,m^{-3}}$ at $0\,°\mathrm{C}$ ($12{,}361\,\mathrm{ppm}$). The air is assumed fully saturated with respect to supercooled liquid.

rated air at $0\,°\mathrm{C}$. These limits correspond to $512\,\mathrm{ppm}$ and $12{,}361\,\mathrm{ppm}$ at standard pressure, respectively. Fully saturated air is assumed, as high relative humidity is typical during measurement in closed circuit icing wind tunnels. Increasing lock-in integration time can be seen to not yield notable performance improvement, as in the range of interest accuracy is dominated by the uncertainty in the calibration humidity.

5   The determined PA hygrometer accuracy is lower than the accuracy specified for NDIR systems providing a similar measurement range (e.g., $1.5\,\%$; LI-COR Inc., 2020). However, because the accuracy of the hygrometer is currently dominated by the accuracy of the humidity generator, it is expected that improvement of saturator temperature stability and temperature measurement, combined with the independent calibration of the HG, will further improve the accuracy of the hygrometer to similar levels.

10   **4   CWC measurement and uncertainty**

Derivation of the cloud condensed water content from the measured TW mole fraction $x_{\mathrm{w,tot}}$ and the ambient air BWV mole fraction $x_{\mathrm{w,a}}$ requires additional input from the instrument's flow measurement, together with input about the icing wind tunnel operating condition. Equations used to derive the actual condensed water content and the corresponding measurement uncertainty from the measured quantities are briefly described in the following subsections.

15   Measurement of the CWC, defined as the mass of condensed water in the form of hydrometeors per volume of air, is accompanied by hydrometeor and air sampling errors introduced by deviations from the ideal and isokinetic sampling at the TW inlet. These errors are corrected by accounting for the actual mass averaged hydrometeor aspiration efficiency of the probe





for the given particle size distribution $\overline{\eta}_{\mathrm{asp}}$ (cf. Appendix B; Belyaev and Levin, 1974):

$$\mathrm{CWC_i} = \overline{\eta}_{\mathrm{asp}}\,\mathrm{CWC} = \frac{\overline{E}}{\mathrm{IKF}}\,\mathrm{CWC}\;. \tag{3}$$

Here, $\mathrm{CWC_i}$ is the indicated or measured condensed water content and $\overline{E}$ is the mass averaged hydrometeor collection efficiency of the probe.

5     Under ideal and isokinetic sampling conditions, the CWC is equal to the ratio of the mass flow rate of hydrometeors to the volumetric flow rate of air entering the probe TW inlet. At the inlet, the volume of air occupied and displaced by the liquid or solid hydrometeors can be assumed negligible for the water contents of interest (Davison et al., 2016). Indicated condensed water content $\mathrm{CWC_i}$ is the ratio of the actually sampled hydrometeor mass flow rate $\dot{m}_{\mathrm{h}}$ to the sampled volumetric flow rate of air $q_{\mathrm{a}}$. Thus, using Eq. (3), CWC may be calculated from the expression

$$\mathrm{CWC} = \mathrm{CWC_i}\,\frac{\mathrm{IKF}}{\overline{E}} = \frac{\dot{m}_{\mathrm{h}}}{q_{\mathrm{a}}}\cdot\frac{\mathrm{IKF}}{\overline{E}}\;. \tag{4}$$

## 4.1   Indicated CWC

The flow rates $\dot{m}_{\mathrm{h}}$ and $q_{\mathrm{a}}$ may be expressed in terms of the total mass flow sampled through the TW inlet $\dot{m}_{\mathrm{tot}}$ (IWT air, including hydrometeors), the mass flow of humid ambient air $\dot{m}_{\mathrm{a}}$ (IWT air, excluding hydrometeors) and the ambient air density $\rho_{\mathrm{a}}$. The indicated CWC is then calculated from

$$\mathrm{CWC_i} = \frac{\dot{m}_{\mathrm{h}}}{q_{\mathrm{a}}} = \frac{\dot{m}_{\mathrm{tot}} - \dot{m}_{\mathrm{a}}}{\dot{m}_{\mathrm{a}}/\rho_{\mathrm{a}}} \tag{5}$$

$$= \rho_{\mathrm{a}}\left(\frac{\omega_{\mathrm{da,a}}}{\omega_{\mathrm{da,tot}}} - 1\right) \tag{6}$$

$$= \frac{p_{\mathrm{a}}\,M_{\mathrm{w}}}{R\,T_{\mathrm{a}}}\cdot\frac{x_{\mathrm{w,tot}} - x_{\mathrm{w,a}}}{1 - x_{\mathrm{w,tot}}}\;, \tag{7}$$

where the density of the air has been calculated assuming an ideal gas mixture of dry air (subscript $\mathrm{da}$) and water vapor,

$$\rho_{\mathrm{a}} = \rho_{\mathrm{da}} + \rho_{\mathrm{w,a}} \tag{8}$$

$$= \frac{p_{\mathrm{a}}}{R\,T_{\mathrm{a}}}\left[M_{\mathrm{da}}(1 - x_{\mathrm{w,a}}) + M_{\mathrm{w}}\,x_{\mathrm{w,a}}\right]\;. \tag{9}$$

$\omega_{\mathrm{da,tot}}$ and $\omega_{\mathrm{da,a}}$ are the dry air mass fractions of the sampled TW air, which includes evaporated hydrometeors, and of the ambient air, respectively:

$$\omega_{\mathrm{da,tot}} = \frac{\dot{m}_{\mathrm{da}}}{\dot{m}_{\mathrm{tot}}} = \frac{M_{\mathrm{da}}(1 - x_{\mathrm{w,tot}})}{M_{\mathrm{da}}(1 - x_{\mathrm{w,tot}}) + M_{\mathrm{w}}\,x_{\mathrm{w,tot}}}\;, \tag{10}$$

$$\omega_{\mathrm{da,a}} = \frac{\dot{m}_{\mathrm{da}}}{\dot{m}_{\mathrm{a}}} = \frac{M_{\mathrm{da}}(1 - x_{\mathrm{w,a}})}{M_{\mathrm{da}}(1 - x_{\mathrm{w,a}}) + M_{\mathrm{w}}\,x_{\mathrm{w,a}}}\;. \tag{11}$$

25   $T_{\mathrm{a}}$ and $p_{\mathrm{a}}$ are the icing wind tunnel static air temperature and pressure. $M_{\mathrm{da}}$ and $M_{\mathrm{w}}$ are the molar masses of dry air and water and $R$ is the universal gas constant. Real gas effects at the measurement temperatures, pressures and humidities of interest are minor.



## 4.2 Isokinetic factor and collection efficiency

The TW inlet flow rate is only set to isokinetic sampling once before activation of the IWT spray system. As the inlet total mass flow rate is held constant and is measured downstream the evaporator, water vapor originating from hydrometeor evaporation reduces the inlet air flow rate during TW measurement, altering the flow field at the probe inlet and reducing the IKF. In addition to this reduction of the IKF, minor changes in the IWT air density $\rho_\mathrm{a}$ or airspeed $U_\mathrm{a}$ during measurement also lead to deviations from the initially set isokineticity.

The isokinetic factor in Eq. (4) corrects for these sources of disproportional sampling of ambient air in comparison to isokinetic sampling and is determined during measurement from

$$\mathrm{IKF} = \frac{\overline{U}_\mathrm{s}}{U_\mathrm{a}} = \frac{\dot{m}_\mathrm{a}}{U_\mathrm{a}\,\rho_\mathrm{a}\,A_\mathrm{inlet}} = \frac{4\,\dot{m}_\mathrm{a}}{U_\mathrm{a}\,\rho_\mathrm{a}\,d_\mathrm{inlet}^2\,\pi}\ , \tag{12}$$

where $d_\mathrm{inlet}$ is the diameter of the circular probe TW inlet.

Since with a decrease of the IKF the collection efficiency at high particle Stokes numbers decreases sub-proportionally to the efficiency at lower Stokes numbers (cf. Fig. 3), condensed water content is overestimated for typical particle size distributions. For each specific particle size distribution encountered during measurement, the mass averaged collection efficiency $\overline{E}$ in Eq. (4) may be used to correct for the size and IKF dependent sampling efficiency.

### 4.2.1 Mass flow measurement

The ambient air mass flow rate $\dot{m}_\mathrm{a}$ required for the calculation of the IKF is determined from the total mass flow sampled through the TW inlet, i.e., the combined mass flow rates through the PA cell $\dot{m}_\mathrm{cell}$ and the bypass path $\dot{m}_\mathrm{bp}$. Together with Eqs. (10)-(11), the ambient air mass flow rate (excluding hydrometeors) through the TW inlet is given by

$$\dot{m}_\mathrm{a} = \frac{\omega_\mathrm{da,tot}}{\omega_\mathrm{da,a}}\,\dot{m}_\mathrm{tot} = \frac{\omega_\mathrm{da,tot}}{\omega_\mathrm{da,a}}\,(\dot{m}_\mathrm{cell} + \dot{m}_\mathrm{bp})\ . \tag{13}$$

The thermal mass flow meters are calibrated for dry air, assuming dry air specific heat capacity for the gas to be measured. As humid air isobaric heat capacity increases by $1\,\%$ at the maximum expected TWC ($10\,\mathrm{g\,m^{-3}}$ CWC, fully saturated air at STP), the indicated volumetric standard flow rates of the flow meters, $q_\mathrm{cell,0}$ and $q_\mathrm{bp,0}$, are converted to humid air mass flow rates (Hardy et al., 1999):

$$\dot{m}_j = \frac{c_\mathrm{p,da}}{c_\mathrm{p,tot}}\,\rho_\mathrm{da,0}\,q_{j,0}$$

$$= \frac{c_\mathrm{p,da}}{c_\mathrm{p,da}\,\omega_\mathrm{da,tot} + c_\mathrm{p,w}\,(1 - \omega_\mathrm{da,tot})}\,\rho_\mathrm{da,0}\,q_{j,0}\ , \tag{14}$$

where $j = \{\mathrm{cell, bp}\}$ refers to the cell or bypass measurement, $\rho_\mathrm{da,0}$ is the dry air density at standard temperature and $1013.25\,\mathrm{hPa}$, $c_\mathrm{p,da}$ is the isobaric specific heat capacity of dry air, and the specific heat capacity of humid air $c_\mathrm{p,tot}$ is calculated assuming an ideal mixture model. The remaining mass flow error after applying the above heat capacity correction has not yet been determined. However, the error is assumed below $1\,\%$, as the change in air specific heat capacity itself is below $1\,\%$ at the maximum expected total water content.





### 4.2.2 CWC estimation

The final expression used for icing wind tunnel CWC estimation is obtained by combining Eq. (4) with Eqs. (7) and (10)-(14):

$$\mathrm{CWC} = \frac{4\,M_{\mathrm{w}}}{\pi\,d_{\mathrm{inlet}}^2\,U_{\mathrm{a}}\,\overline{E}}$$

$$\cdot \frac{\rho_{\mathrm{da},0}\,c_{\mathrm{p,da}}\,(q_{\mathrm{bp},0} + q_{\mathrm{cell},0})}{c_{\mathrm{p,da}}\,M_{\mathrm{da}}\,(1 - x_{\mathrm{w,tot}}) + c_{\mathrm{p,w}}\,M_{\mathrm{w}}\,x_{\mathrm{w,tot}}}$$

$$\cdot \frac{x_{\mathrm{w,tot}} - x_{\mathrm{w,a}}}{1 - x_{\mathrm{w,a}}} \,. \tag{15}$$

Although IWT static air temperature and pressure are required to set the total sampling mass flow to isokinetic TW sampling, this result shows that if the isokinetic factor is not calculated explicitly, air temperature and pressure only appear in the hydrometeor collection efficiency (through air viscosity and slip correction) and otherwise are not required to calculate the condensed water content. For minor temperature and pressure fluctuations during IWT water content measurement, only marginal impact on CWC measurement and uncertainty is anticipated by disregarding changes in IWT air temperature and pressure.

### 4.3 CWC measurement uncertainty

Corrections and errors introduced by the collection efficiency are specific to the respective wind tunnel icing conditions and hence are not considered in the following general analysis. Instead, a mean mass averaged collection efficiency of one is assumed. With the numerically determined collection efficiency given in Fig. 3, maintaining this assumption for the evaluation of the presented measurements in icing conditions of freezing drizzle or rain, with median volume diameters (MVDs) in the range of $100\,\mu\mathrm{m}$ to $650\,\mu\mathrm{m}$, the potential CWC underestimation is below $1\,\%$ (size distribution data taken from Cober et al. (2009)).

The uncertainty of the condensed water content measurement is derived from a first-order propagation of the uncertainties of the quantities appearing in Eq. (15) according to the Guide to the Expression of Uncertainty in Measurement (GUM; Joint Committee for Guides in Metrology, 2008a). Uncertainties not distributed normally have been converted to standard uncertainties for the analytical calculations. Unless otherwise stated, all uncertainties are given in terms of the $95\,\%$ coverage interval. A summary of the individual uncertainties of the input quantities is given in Table C1 in Appendix C of this work.

The current single-hygrometer instrument only allows either TW or BWV content measurement. Alternating between both measurements to determine the CWC inevitably results in a measurement error due to the dynamic behavior of the background water content, which is mainly defined by the initial IWT air saturation level and stability of the temperature conditioning during the measurement. Depending on the saturation level preceding activation of the spray, the background water content during the probe intercomparison increased by up to $0.5\,\mathrm{g\,m}^{-3}$ for as long as five minutes after activation of the spray and before reaching a stable reading. As a consequence of alternating TW and BWV measurement, errors highly depend on subjective assessment during evaluation and are specific to the IWT operating conditions. With the goal of assessing instrument accuracy with a planned second dedicated PA cell for background humidity measurement, the uncertainties of the TW and BWV content measurement are both taken equal to the hygrometer measurement uncertainty given in Section 3.3. The presented uncertainties





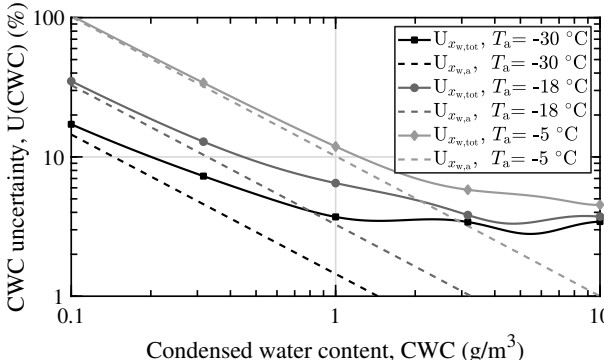

**Figure 8.** Hygrometer measurement uncertainty contributions to the $95\,\%$ CWC measurement uncertainty at three static air temperatures, an airspeed of $60\,\mathrm{m\,s^{-1}}$ and a static air pressure of $1013.25\,\mathrm{hPa}$. Condensed water content uncertainty contributions are given relative to the actual CWC and for isokinetic sampling. The ambient air is assumed fully saturated with respect to supercooled liquid.

may, however, be taken as upper limits for a different hygrometer used for background humidity measurement with similar or better accuracy.

Figure 8 shows the calculated hygrometer contribution to the condensed water content measurement uncertainty at three IWT static air temperatures. Temperatures of $-30\,°\mathrm{C}$, $-18\,°\mathrm{C}$ and $-5\,°\mathrm{C}$ were examined, again assuming fully saturated air

with respect to supercooled liquid water, as this is expected for the closed circuit icing wind tunnel. The measurement uncertainty contributions are given relative to the actual CWC. Contributions of the measurement of the background water vapor concentration ($\mathrm{U}_{x_{\mathrm{w,a}}}$, dashed lines) indicate constant background humidities with associated constant absolute measurement uncertainties.

The hygrometer's contribution to the CWC measurement uncertainty increases rapidly with lower water content and in-

creasing temperature. The latter circumstance is a result of the rising absolute BWV concentration uncertainty with increasing background humidity, which dominates the difference of measured total and background water vapor concentrations at low CWCs (last term in Eq. (15)). For a condensed water content of $0.5\,\mathrm{g\,m^{-3}}$ and an IWT temperature of $-5\,°\mathrm{C}$, the combined hygrometer uncertainty contribution (root of sum of squares) is $0.15\,\mathrm{g\,m^{-3}}$. At $-30\,°\mathrm{C}$ the hygrometer's contribution is reduced to $0.03\,\mathrm{g\,m^{-3}}$.

Figure 9 shows the overall CWC measurement uncertainty at two of the above temperatures. Also shown are the individual contributions of the input quantities. At high condensed water contents the device is currently obviously limited by the large relative uncertainty in the probe TW inlet area ($\pm 9\,\%$), which contributes a constant $10.5\,\%$ to the overall uncertainty. This is a result of the particularly small size of the TW inlet diameter. However, deviation of the estimated nozzle inlet area from the true size only results in an invariant systematic error in the isokinetic factor. Hence the CWC measurement error should

be proportional to the indicated CWC and should not affect instrument precision. The additional error in the projected probe TW inlet area due to misalignment to the direction of flow is below $0.5\,\%$, assuming an angle of attack within $\pm 5\,°$. This does



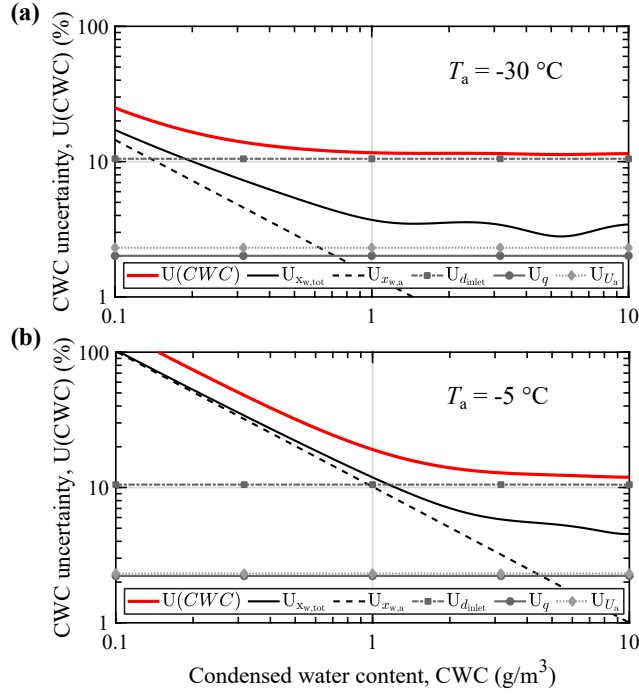

**Figure 9.** Condensed water content measurement uncertainty ($95\%$) and individual contributions at static air temperatures of **(a)** $-30\,°\mathrm{C}$ and **(b)** $-5\,°\mathrm{C}$. Uncertainties are given relative to the actual CWC and for isokinetic sampling. Wind speed and static air pressure are $60\,\mathrm{m\,s^{-1}}$ and $1013.25\,\mathrm{hPa}$, respectively. The ambient air is assumed fully saturated with respect to supercooled liquid.

not include errors induced by changes in collection efficiency, which again have to be considered separately for the respective particle size distribution.

Towards lower condensed water contents, the overall measurement uncertainty is dominated by the humidity measurement. As is typical for IKPs, the accuracy of the instrument is highest at low ambient temperatures or background humidities (Davison

et al., 2016). Table 1 summarizes absolute and relative measurement uncertainties $\mathrm{U(CWC)}$ at static air temperatures of $-5\,°\mathrm{C}$ and $-30\,°\mathrm{C}$ for condensed water contents in the range of $0.25\,\mathrm{g\,m^{-3}}$ to $3\,\mathrm{g\,m^{-3}}$.

At the lower temperature, measurement uncertainty decreases below $20\%$ above a condensed water content of $0.14\,\mathrm{g\,m^{-3}}$. In warm air of $-5\,°\mathrm{C}$ this is only the case above a CWC of $0.93\,\mathrm{g\,m^{-3}}$. Due to the high contributions of humidity measurement and inlet area uncertainty, the measurements of the total mass flow and IWT airspeed only marginally contribute to the overall

uncertainty.

To validate the stated first-order analytic CWC uncertainties, a Monte Carlo method (Joint Committee for Guides in Metrology, 2008b) has been applied. The method takes into account and propagates the assumed uncertainty distributions of the input quantities. As the TW inlet diameter is assumed with uniform probability within the measured and specified bounds (cf. Table C1), analytic and numeric uncertainties are expected to differ at high CWCs where the inlet diameter contribution dom-

15 inates. Numerically calculated shortest $95\%$ coverage intervals attained with the Monte Carlo method lie within the analytic





**Table 1.** Instrument absolute and relative CWC measurement uncertainties (95 % coverage) at selected cloud CWCs and at static air temperatures of $-5\,°C$ and $-30\,°C$. Wind tunnel airspeed and static air pressure for the calculations are $60\,\mathrm{m\,s^{-1}}$ and $1013.25\,\mathrm{hPa}$, respectively.

| $T_\mathrm{a}$ | CWC | U(CWC) | |
|---|---|---|---|
| $-5\,°C$ | $0.25\,\mathrm{g\,m^{-3}}$ | $0.15\,\mathrm{g\,m^{-3}}$ | 60 % |
| | $0.50\,\mathrm{g\,m^{-3}}$ | $0.16\,\mathrm{g\,m^{-3}}$ | 31 % |
| | $1.00\,\mathrm{g\,m^{-3}}$ | $0.19\,\mathrm{g\,m^{-3}}$ | 19 % |
| | $3.00\,\mathrm{g\,m^{-3}}$ | $0.39\,\mathrm{g\,m^{-3}}$ | 13 % |
| $-30\,°C$ | $0.25\,\mathrm{g\,m^{-3}}$ | $0.04\,\mathrm{g\,m^{-3}}$ | 15 % |
| | $0.50\,\mathrm{g\,m^{-3}}$ | $0.06\,\mathrm{g\,m^{-3}}$ | 13 % |
| | $1.00\,\mathrm{g\,m^{-3}}$ | $0.12\,\mathrm{g\,m^{-3}}$ | 12 % |
| | $3.00\,\mathrm{g\,m^{-3}}$ | $0.34\,\mathrm{g\,m^{-3}}$ | 11 % |

interval over the whole range of interest (cf. Appendix E, Fig. E1). Hence, the presented analytic uncertainties may be taken as upper bounds to a more realistic estimation of the uncertainty.

## 5 Icing wind tunnel probe intercomparison

The photoacoustic hygrometer in combination with the IKP was used for TW measurement during a water content probe intercomparison campaign at the RTA Rail Tech Arsenal Fahrzeugversuchsanlage GmbH (RTA) icing wind tunnel. The closed circuit IWT is capable of simulating air temperatures down to $-30\,°C$ and windspeeds up to $80\,\mathrm{m\,s^{-1}}$ in a test section of $(3.5{\times}2.5{\times}3)\,\mathrm{m}$ (width×height×length) at local ambient pressure. Test conditions included freezing drizzle and rain icing conditions with bi-modal particle size distributions (in close agreement to EASA CS-25 Appendix O) and with MVDs of approximately $100\,\mathrm{\mu m}$ and $550\,\mathrm{\mu m}$ to $650\,\mathrm{\mu m}$, respectively. Measurements in classical supercooled droplet icing conditions at higher cloud CWCs had to be disregarded due to the already described oscillations observed in the TW measurements at high loads, suspected to be caused by temporary obstructions of the small diameter inlet. All measurements were conducted at a target static air temperature of $-5\,°C$ and wind speed of $60\,\mathrm{m\,s^{-1}}$. Freezing drizzle is created by 264 pneumatic atomizing nozzles mounted on horizontal spray bars placed approximately $12\,\mathrm{m}$ upstream the test section. Freezing rain droplet size distributions with maximum diameters of approximately $1.5\,\mathrm{mm}$ were generated with an additional set of twelve rotating nozzles mounted on the IWT spray bar system (cf. Breitfuss et al., 2019).

The PA system was compared against a multi-element water content hot-wire probe (SEA, WCM-2000 Multi Element Water Content System) and an IKP from Cranfield University (CU-IKP; Bansmer et al., 2018), which utilizes commercial NDIR sensor based hygrometers for simultaneous BWV and isokinetic TW measurement via backward- and forward-facing inlets and was specifically designed for high water content IWT measurement. All probes were mounted side by side on a horizontal splitter-plate-like panel with the probe inlets positioned at the approximate vertical center of the test section (cf. Fig. 10) and



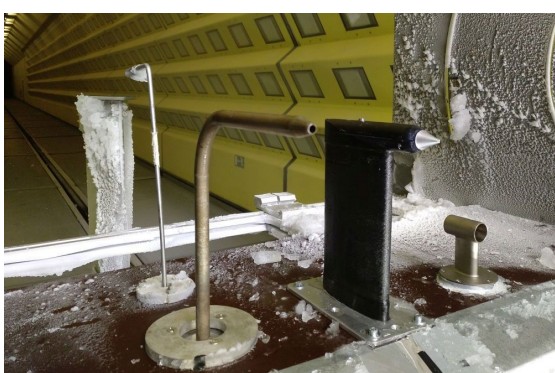

**Figure 10.** Positioning of the water content probes mounted on the splitter-plate-like panel in the RTA icing wind tunnel test section. Viewing direction is in the direction of flow. From left to right: Cranfield University IKP TW and backward facing BWV inlet, PA hygrometer IKP and SEA WCM-2000.

the measurements with the probes were conducted simultaneously. In the relevant area of the test section, LWC spatial cloud uniformity of the IWT is assumed better than $\pm 10\,\%$ and $\pm 15\,\%$ for freezing drizzle and freezing rain, respectively. The spatial cloud uniformity was determined with an icing cloud calibration grid (Breitfuss et al., 2019) and is within the SAE ARP-5905 recommended maximum allowable deviation ($\pm 20\,\%$).

During the intercomparison the PA hygrometer was primarily used for TWC measurement. Continuous background humidity measurement was thus performed with an external capacitive humidity sensor (E+E Elektronik, EE33) mounted to the IWT wall. The sensor (labeled *IWT humidity rear*) has a specified relative humidity and temperature measurement accuracy better than $\pm 2.3\,\%\mathrm{RH}$ and $\pm 0.25\,^{\circ}\mathrm{C}$, respectively, and is located downstream the PA system IKP at the rear end of the IWT. BWV concentrations measured by this sensor were time-shifted to correct for the time delay resulting from the displacement from

the probe location. A second humidity sensor of the same type (*IWT humidity front*) is placed at the test section, but is not directly exposed to the main IWT air flow. This sensor is not used for evaluation, but gives an indication of the true background humidity at the sampling point of the hygrometer.

Figures 11 and 12 show two measurements in freezing rain with a drop MVD of approximately $550\,\mu\mathrm{m}$. The upper panels show the TW and BWV mole fractions measured by the PA system and the CU-IKP together with background humidities

measured by both IWT humidity sensors over time. The lower panels of Figs. 11 and 12 show the corresponding derived CWC for the PA system and the CU-IKP, as well as the measured CWC by the multi-element hot-wire instrument. Activation of the IWT spray system is indicated by a calculated theoretical condensed water content (*IWT spray*), which, however, is known to underestimate the true CWC in SLD icing conditions. The high dispersion in the PA signal during cloud measurement is a result of the low averaging effect of the small probe TW inlet area in combination with the fast response time of the hygrometer

($\tau_{63} < 2\,\mathrm{s}$). Collection efficiency has been assumed $100\,\%$ for the evaluation, as the error is assumed below $1\,\%$ for the SLD size distributions (cf. Section 4.3).

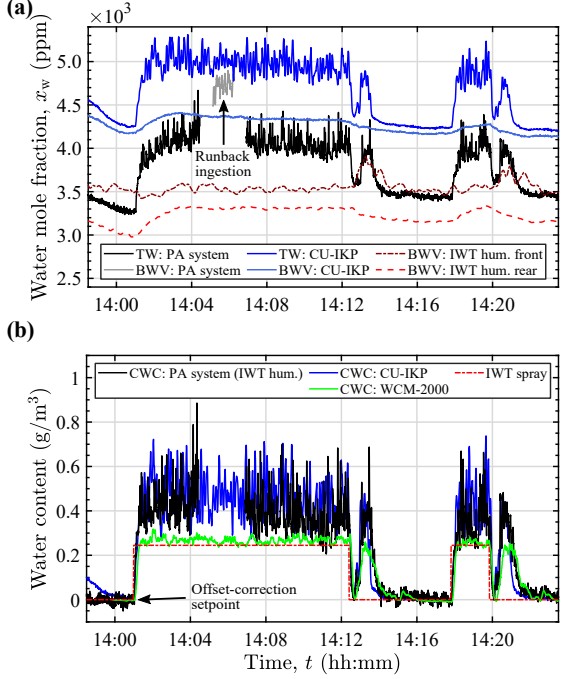

**Figure 11.** Water content measurements in freezing rain with a drop MVD of $550\,\mu m$ at $-5\,°C$ and $60\,m\,s^{-1}$. **(a)** PA instrument and CU-IKP TW and BWV mole fractions (erroneous PA instrument BWV sampling perpendicular to flow), together with BWV mole fractions calculated from the IWT humidity measurements. **(b)** CWCs determined by the PA system (in combination with *IWT humidity rear*), the CU-IKP and the hot-wire probe (WCM-2000). Spray activation is indicated by *IWT spray*.

The external background humidity reference (*IWT humidity rear*) can be seen to correlate well with the PA system total water measurement when the spray system is inactive (cloud-free air). Nevertheless, considerable offset (several hundred ppm) was measured in all conditions and was therefore subtracted for the estimation of the condensed water content. Points in time of the $10\,s$ offset calculation period are indicated with arrows in Figs. 11(b) and 12(b). The observable offset is mainly attributed

5    to the humidity sensor accuracy, as well as to gradients in the IWT air temperature and saturation between the measurement locations.

The CU-IKP likewise indicated a steady offset of approximately $100\,ppm$ between the TW and BWV measurements when the spray system was inactive. This difference may have resulted from differing sensitivities or zero offset drift of the hygrometer channels and is corrected in a similar manner as with the PA system. Additionally, as the CU-IKP has not been calibrated

10    for absolute measurement and the NDIR gas analyzer has been used without continuous reference measurement, exhibiting simultaneous but similar drift of both channels, measured concentrations were larger than determined by the IWT humidity sensors and the PA system. For some measurements the difference between PA system and CU-IKP exceeded $2,000\,ppm$. As large parts of humidities measured by the CU-IKP are in excess of the saturation mole fraction with respect to supercooled liquid (even well before activation of the spray), it is concluded that the calculated CU-IKP values overestimate true absolute

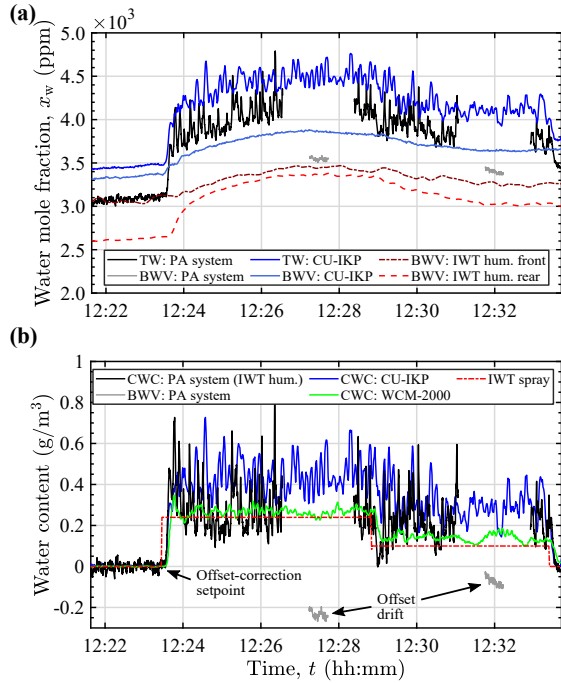

**Figure 12.** Water content measurement in freezing rain (MVD of $550\,\mu m$, $-5\,^{\circ}C$, $60\,m\,s^{-1}$), showing underestimated CWC due to significant background humidity offset drift. **(a)** PA instrument and CU-IKP TW and BWV mole fraction (extended PA instrument probe BWV inlet), together with BWV mole fractions calculated from the IWT humidity measurements. **(b)** CWCs determined by the PA system (in combination with *IWT humidity rear*), the CU-IKP and the hot-wire probe (WCM-2000). *BWV: PA system* shows the residual background offset between the Pa system's BWV measurement and *IWT humidity rear* after offset correction. Spray activation is indicated by *IWT spray*.

TW and BWV contents. Effects on CWC derivation, however, are mitigated by the expected similar drift of both channels and the primarily differential nature of CWC measurement.

Background humidity measurement with the PA instrument's BWV inlet port oriented perpendicular to the direction of flow resulted in highly elevated BWV levels (Fig. 11(a)), due to ingestion of runback water or sampling of air from the humidified
5   thermal boundary layer of the heated probe. Therefore, the latter half of the measurements was conducted with the probe BWV inlet extended by a backward-oriented tubing, which enabled intermittent and more reliable background humidity measurement in icing conditions. Differences (residuals) in background humidities measured by the PA system with the modified BWV inlet and the reference humidity sensor were used to identify measurements exhibiting considerable background humidity offset drift (cf. Fig. 12(a) and (b)), which were subsequently excluded from further evaluation. Due to the dynamic behavior of the
10  background humidity, estimated offset drifts of up to $0.1\,g\,m^{-3}$ could not be reliably detected with the described method and may have resulted in equivalent CWC measurement errors. For the water contents encountered during the intercomparison this may have resulted in relative errors of $11\,\%$ to $36\,\%$ for the highest and the lowest CWC, respectively.





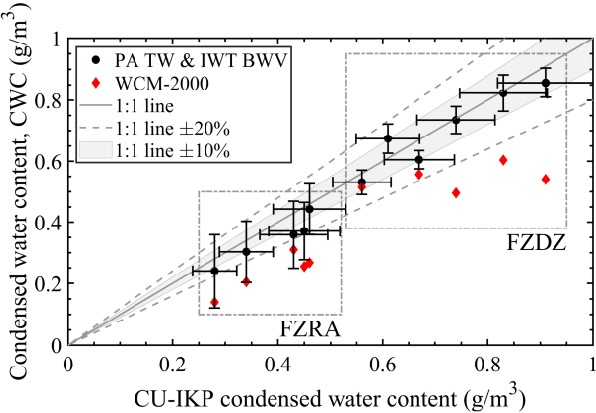

**Figure 13.** Mean measured CWC of the PA system in combination with the IWT background humidity measurement and of the hot-wire probe (WCM-2000) over the CWC measured by the Cranfield University IKP (CU-IKP). Dash-dotted rectangular boxes in the figure mark measurements of freezing rain (FZRA) and freezing drizzle (FZDZ) with MVDs of approximately $550\,\mu m$ to $650\,\mu m$ and $100\,\mu m$, respectively. Vertical error bars mark the standard deviations of the measurements. Horizontal error bars indicate the IWT cloud LWC uniformity ($\pm 15\,\%$ and $\pm 10\,\%$ for freezing drizzle and freezing rain, respectively).

Figure 13 finally shows the comparison of the mean CWCs measured by all probes in conditions of freezing drizzle and freezing rain. Condensed water contents determined with the PA system and the hot-wire probe are plotted over the mean CWC measured by the Cranfield University IKP, as the device has been assumed the reference due to its superior probe design and the simultaneous background and TW measurement.

Precision of the presented measurements heavily depends on the stability of the background humidity during total water content measurement with the PA system and the correct identification of background humidity drifts. Although the measurement uncertainty cannot be quantified for the applied method of BWV estimation at the location of the IKP, the CWC derived from the PA system TW measurement is shown to agree within $\pm 20\,\%$ of the reference measurement for conditions of freezing drizzle or rain. Condensed water contents determined in freezing drizzle are within $\pm 10\,\%$ of the reference (best-fit slope of
0.98).

Total condensed water content determined by the hot-wire instrument continuously was below the CWC measured by the CU-IKP. This underestimation is partly attributed to splashing of large droplets from the hot-wire sensor-element (cylindrical half-pipe facing in the direction of flow), but is larger than is anticipated for freezing drizzle cloud droplet distributions (Steen et al., 2016) and may indicate the advantages of isokinetic evaporator probes for CWC measurement in these conditions. A
detailed analysis of the severe differences is outside the scope of this work. However, measured deviations from the reference IKP may also be attributable to spatial IWT cloud non-uniformity for all systems. Accuracy of PA system CWC measurement is additionally decreased by the high absolute background humidity present at the relatively warm IWT temperature during the measurements (cf. Section 4.3).


## 6    Conclusions and outlook

In this work a hygrometer based on intensity-modulated photoacoustic spectroscopy with a near-infrared laser diode has been realized and combined with a two-pressure humidity generator and an isokinetic evaporator probe to provide a new instrument capable of measuring total or background water contents in simulated atmospheric icing conditions. The dynamic range of

the single-wavelength PA hygrometer has been shown to encompass water contents occurring in SLD, mixed-phase and high IWC environments, where classical water content probes are associated with lower accuracy. Laboratory calibration of the hygrometer using the instrument's calibration unit displayed a $1\,\text{s}$ integration time limit of detection of $23\,\text{ppm}$ and an accuracy (95 % coverage) better than 2.5 % to 3.3 % in the range of $512\,\text{ppm}$ to $12,361\,\text{ppm}$ at standard pressure. The range corresponds to a saturated sea-level cloud-free air at $-30\,^{\circ}\text{C}$ and a CWC of $5\,\text{g}\,\text{m}^{-3}$ in saturated $0\,^{\circ}\text{C}$ air. Since the determined accuracy

is dominated by the uncertainty of the built-in humidity reference, further improvement of the hygrometer's measurement uncertainty may be achieved by using an independent traceable calibration.

For CWC measurements a major contribution to the overall measurement uncertainty is associated with the small diameter TW inlet of the IKP ($3.3\,\text{mm}$), which currently constrains the device uncertainty (95 % coverage) to above 10 % in all conditions. The small diameter also is suspected to cause temporary accumulation of water or ice in the inlet at high CWC loads. To

further decrease the overall measurement uncertainty to the level of the hygrometer uncertainty, a redesign of the IKP inlet is the focus of ongoing research. The isokinetic aspiration efficiency of the probe at wind tunnel airspeeds above $60\,\text{m}\,\text{s}^{-1}$ has been determined by numerical means and near standard pressures and has been shown to lie above 88 % for droplets of any size and above 99 % for droplets with diameters greater than $40\,\mu\text{m}$. From the determined size dependent collection efficiency a bias of less than 1 % can be inferred for CWC measurement in absence of detailed droplet size distribution data in conditions

of freezing drizzle or rain (EASA CS-25 Appendix O).

Uncertainty considerations showed that despite the current limitations given by the IKP inlet, an accuracy better than 20 % is achieved by the instrument for CWCs above $0.14\,\text{g}\,\text{m}^{-3}$ in cold air ($-30\,^{\circ}\text{C}$) and when combined with a suitable background humidity measurement. For higher condensed water contents measurement accuracy further improves. In saturated warm air ($-5\,^{\circ}\text{C}$) the hygrometer uncertainty currently limits practical measurement to condensed water contents above $0.9\,\text{g}\,\text{m}^{-3}$. With

additional adaptations of the TW inlet and improvement of the calibration process, further extension of the useful operating range to lower water contents is expected. It has to be noted that the determined measurement uncertainty is higher than the $\pm 10\,\%$ LWC measurement instrumentation maximum uncertainty demanded by the SAE ARP-5905, which, however, has been defined for classical icing conditions (EASA CS-25 Appendix C) and may be increased in a similar recommended practice for the particularly challenging measurement in SLD icing conditions (SAE AIR-6341, 2015).

The system's TWC measurement capability has been deployed in a CWC measurement intercomparison with a reference IKP instrument in freezing drizzle and rain conditions in the RTA icing wind tunnel. Background humidity had to be estimated independently by an external humidity sensor which, together with the necessary method of offset correction, was determined to limit achievable measurement precision for the chosen setup. Measurements performed in warm air freezing drizzle and rain conditions with MVDs from $100\,\mu\text{m}$ to $650\,\mu\text{m}$, however, showed a CWC agreement of the two IKPs within $\pm 20\,\%$ for





water contents in the range of $0.3\,\mathrm{g\,m^{-3}}$ to $0.9\,\mathrm{g\,m^{-3}}$. This is also within the recommended maximum LWC spatial deviation allowed by the SAE ARP-5905 ($\pm 20\,\%$).

## Appendix A: Photoacoustic background signal correction

The signal returned by the PA hygrometer is the lock-in signal $\boldsymbol{S_m} = (S_{m,I}, S_{m,Q})^T$, where $I$ and $Q$ denote the in-phase and quadrature components of the lock-in amplifier, respectively. Prior to each calibration, a background photoacoustic signal, $\boldsymbol{S}_{BG} = (S_{BG,I}, S_{BG,Q})^T$ is recorded after flushing the PA cell with the zero air until a stable reading is attained.

The photoacoustic amplitude of all subsequent calibration or water content measurements is calculated on the digital signal processing unit of the hygrometer after phase-correct subtraction of the mean of the recorded PA background signal:

$$
\begin{aligned}
S &= \|\boldsymbol{S}_m - \overline{\boldsymbol{S}}_{BG}\| \\
&= \sqrt{(S_{m,I} - \overline{S}_{BG,I})^2 + (S_{m,Q} - \overline{S}_{BG,Q})^2}\ .
\end{aligned}
\tag{A1}
$$

## Appendix B: Calculation of inlet efficiencies, Stokes number and stopping distance

The aspiration efficiency $\eta_{\mathrm{asp}}$ of particles at a given particle size $d_{\mathrm{p}}$ is given by the particle mass concentration in the air entering the inlet divided by the ambient mass concentration at that size (Belyaev and Levin, 1974),

$$
\eta_{\mathrm{asp}}(d_{\mathrm{p}}) = \frac{\mathrm{CWC_i}(d_{\mathrm{p}})}{\mathrm{CWC}(d_{\mathrm{p}})}\ ,
\tag{B1}
$$

and may be written in terms of the limiting area $A_{\mathrm{lim}}$ in front of the inlet, within which all trajectories of sampled particles begin, and the freestream to mean sampling velocity ratio $U_{\mathrm{a}}/\overline{U}_{\mathrm{s}}$:

$$
\eta_{\mathrm{asp}}(d_{\mathrm{p}}) = \frac{A_{\mathrm{lim}}(d_{\mathrm{p}})}{A_{\mathrm{inlet}}} \cdot \frac{U_{\mathrm{a}}}{\overline{U}_{\mathrm{s}}} = \frac{E(d_{\mathrm{p}})}{\mathrm{IKF}}\ .
\tag{B2}
$$

Here, $E = A_{\mathrm{lim}}/A_{\mathrm{inlet}}$ is the particle size dependent collection efficiency and $\mathrm{IKF} = \overline{U}_{\mathrm{s}}/U_{\mathrm{a}}$ is the isokinetic factor.

For the evaluation of the collection efficiencies, the particle Stokes number $St_{\mathrm{p}}$ is calculated according to Kulkarni et al. (2011):

$$
St_{\mathrm{p}} = \frac{\rho_{\mathrm{p}}\, d_{\mathrm{p}}^2\, U_{\mathrm{p}}\, C_c}{18\, \eta\, d_{\mathrm{inlet}}}\ ,
\tag{B3}
$$

where $\rho_{\mathrm{p}}$ is the droplet density calculated for supercooled liquid water (Hare and Sorensen, 1987), $d_{\mathrm{p}}$ is the droplet diameter, $U_{\mathrm{p}}$ is the initial droplet velocity equal to the freestream airspeed $U_{\mathrm{a}}$, $C_c$ is the Cunningham slip correction, $\eta$ is the air dynamic viscosity and $d_{\mathrm{inlet}}$ is the probe inlet diameter.

The Cunningham slip correction for droplets is calculated by

$$
C_c = 1 + \frac{2\,\lambda}{d_{\mathrm{p}}}\left[1.207 + 0.440 \exp\left(-0.596\, d_{\mathrm{p}}/(2\,\lambda)\right)\right]
\tag{B4}
$$



**Table C1.** Summary of uncertainties of the two-pressure humidity generator (HG) and the input quantities entering the condensed water content calculation. Uncertainties are given in terms of half-widths of the rectangular uncertainty distributions.

| Variable | Description | Reference / Source | Uncertainty |
|---|---|---|---|
| $x_{\mathrm{w,BG}}$ | Zero air residual water vapor volume fraction | calibrated | 2 ppmv |
| $T_{\mathrm{HG}}$ | HG saturator air temperature | calibrated | 0.16 K |
| $p_{\mathrm{HG}}$ | HG saturator air pressure | calibrated | 200 Pa (2 %FS) |
| $M_{\mathrm{w}}$ | Molar mass of water ($18.01528\,\mathrm{g\,mol^{-1}}$) | Wieser and Berglund (2009) | negligible |
| $M_{\mathrm{da}}$ | Molar mass of dry air ($28.964\,\mathrm{g\,mol^{-1}}$) | Giacomo (1982) | negligible |
| $c_{\mathrm{p,w}}$ | Water vapor specific heat capacity ($1874\,\mathrm{J\,kg^{-1}\,K^{-1}}$, $x_{\mathrm{w}} = 1\,\%$) | Bell et al. (2014) | negligible |
| $c_{\mathrm{p,da}}$ | Dry air specific heat capacity ($1006.7\,\mathrm{J\,kg^{-1}\,K^{-1}}$) | Bell et al. (2014) | negligible |
| $\rho_{\mathrm{da,0}}$ | Dry air density at $0\,^{\circ}\mathrm{C}$ and 1013.25 hPa ($1.293\,\mathrm{kg\,m^{-3}}$) | Giacomo (1982) | negligible |
| $q_{\mathrm{bp,0}}$ | Bypass path standard volumetric flow rate | calibrated | 0.06 slpm (1 %FS) |
| $q_{\mathrm{cell,0}}$ | PA cell standard volumetric flow rate | calibrated | 0.6 slpm (1 %FS) |
| $d_{\mathrm{inlet}}$ | Probe TW inlet diameter | measurement | 0.15 mm |
| $U_{\mathrm{a}}$ | Wind tunnel airspeed at probe | IWT | 2 % |

(Allen and Raabe, 1985; Rader, 1990), where the mean free path $\lambda$ according to Willeke (1976) is given by:

$$\lambda = \lambda_r \left(\frac{101 \cdot 10^3}{p_{\mathrm{a}}}\right) \left(\frac{T_{\mathrm{a}}}{293}\right) \left(\frac{1 + 101/293}{1 + 101/T_{\mathrm{a}}}\right) . \tag{B5}$$

The air dynamic viscosity is calculated by

$$\eta = \eta_r \left(\frac{T_r + S_u}{T_{\mathrm{a}} + S_u}\right) \left(\frac{T_{\mathrm{a}}}{T_r}\right)^{3/2} \tag{B6}$$

(Kulkarni et al., 2011), where the reference viscosity $\eta_r$ is $18.33 \times 10^{-6}\,\mathrm{Pa\,s}$ and the Sutherland interpolation constant $S_u$ is 110.4 K at the reference temperature $T_r$ of 293 K.

The particle stopping distance $S_{\mathrm{p}}$ for droplet or particle Reynolds numbers $Re_{\mathrm{p}}$ in the range of 1 to 400 is calculated with the correlation obtained by Mercer (1973):

$$S_{\mathrm{p}} = \frac{\rho_{\mathrm{p}}\,d_{\mathrm{p}}}{\rho_{\mathrm{a}}} \left(Re_{\mathrm{p}}^{1/3} - \sqrt{6}\,\mathrm{atan}\left(\frac{Re_{\mathrm{p}}^{2/3}}{\sqrt{6}}\right)\right) . \tag{B7}$$

## Appendix C: Summary of input uncertainties

Table C1 summarizes individual uncertainty contributions to the overall instrument CWC measurement uncertainty.





## Appendix D: Hygrometer uncertainty

The theoretical background corrected lock-in signal amplitude for a given water vapor mole fraction in air $x_\text{w}$ and a parameter set $\boldsymbol{b}$ may be written as $S = f(x_\text{w}, \boldsymbol{b})$ (Lang et al., 2020). To determine the parameters in the calibration function with the least-squares method (Eq. (2)) while considering the calibration humidity uncertainty $u(x_{\text{w},i})$, the inverse function $x_\text{w} = f^{-1}(S, \boldsymbol{b})$ is required. As no closed-form expression for $x_\text{w}$ can be found, the water vapor mole fraction is obtained by numerically finding the root of

$$g(S, x_\text{w}, \boldsymbol{b}) = S - f(x_\text{w}, \boldsymbol{b}) \tag{D1}$$

for a measured signal amplitude and a given set of parameters:

$$x_\text{w} = f^{-1}(S, \boldsymbol{b}) = \{x \,|\, g(S, x, \boldsymbol{b}) = 0\} \ . \tag{D2}$$

The measurement uncertainty of the PA hygrometer $u(x_\text{w})$ is then evaluated from Eq. (D2) by combining the uncertainties of the measurement signal amplitude $u(S)$ and the correlated parameters determined from calibration, following the GUM (Joint Committee for Guides in Metrology, 2008a):

$$u^2(x_\text{w}) = \left(\frac{\partial f^{-1}}{\partial S}\right)^2 u^2(S)$$
$$+ \sum_{i=1}^{5}\sum_{j=1}^{5} \frac{\partial f^{-1}}{\partial b_i} \frac{\partial f^{-1}}{\partial b_j} u(b_i, b_j) \ , \tag{D3}$$

where $u(b_i, b_j)$ is the covariance of the fit parameters $b_i$ and $b_j$. $u(b_i, b_i) = u^2(b_i)$ is the variance of coefficient $b_i$.

The uncertainty in the measured signal amplitude is estimated from the Allan deviation analysis and is taken equivalent to the signal noise at the measurement integration time of $1\,\text{s}$.

The sensitivity coefficients in Eq. (D3), i.e., the partial derivatives of $f^{-1}$ with respect to the PA signal amplitude and the calibration function parameters, are calculated from Eqs. (D1) and (D2) by using standard rules of calculus (Lira, 2002):

$$\frac{\partial f^{-1}}{\partial S} = -\frac{\partial g/\partial S}{\partial g/\partial x_\text{w}} = \frac{1}{\partial f/\partial x_\text{w}} \ , \tag{D4}$$
$$\frac{\partial f^{-1}}{\partial b_i} = -\frac{\partial g/\partial b_i}{\partial g/\partial x_\text{w}} = -\frac{\partial f/\partial b_i}{\partial f/\partial x_\text{w}} \ . \tag{D5}$$

Errors introduced by finding the root in Eq. (D2) are assumed negligible, due to the high accuracy of the numerical solver with the chosen tolerance level.

## Appendix E: Numerical CWC uncertainty evaluation

Figure E1 shows the comparison of the $95\,\%$ coverage intervals of the CWC measurement uncertainty calculated with the first-order analytical and the Monte Carlo method for an IWT static air temperature of $-30\,°\text{C}$ and an airspeed of $60\,\text{m}\,\text{s}^{-1}$.





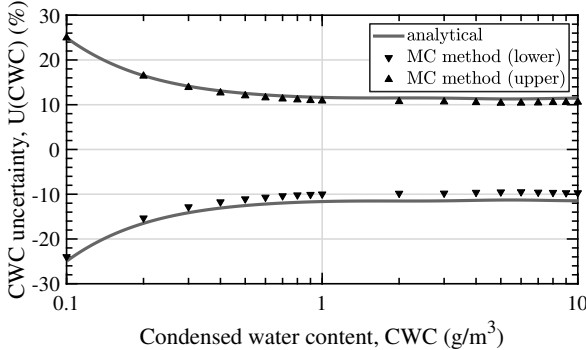

**Figure E1.** 95 % coverage intervals of the CWC measurement uncertainty calculated with the first-order analytical and the Monte Carlo (MC) method. Uncertainties are given relative to the actual CWC. Icing wind tunnel static air temperature, pressure and airspeed are set to $-30\,°C$, $1013.25\,hPa$ and $60\,m\,s^{-1}$, respectively, and the ambient air is assumed fully saturated with respect to supercooled liquid water. Monte Carlo intervals are shortest (non-symmetric) 95 % intervals indicated by the lower and upper bounds.

Uncertainties are given relative to the actual CWC. Shortest intervals obtained by the Monte Carlo method can be seen to lie within the analytical intervals over the whole CWC range of interest.

*Author contributions.* Conceptualization: B. Lang, W. Breitfuss, W. Hassler, A. Bergmann; Methodology and investigation: B. Lang, P. Breitegger, W. Breitfuss, A. Tramposch, H. Pervier; Software and material preparation: B. Lang, W. Breitfuss, S. Schweighart, H. Pervier; Writing - original draft preparation: B. Lang; Writing - review and editing: all authors; Funding acquisition: A. Tramposch, A. Bergmann, W. Hassler; Resources: A. Bergmann, A. Klug, W. Hassler; Supervision: A. Bergmann, W. Hassler; Project administration: A. Bergmann, A. Klug, W. Hassler.

*Competing interests.* The authors declare that they have no conflict of interest.

*Acknowledgements.* This work has been partly funded by the Austrian Aeronautics Programme TAKE OFF of the Federal Ministry of Transport, Innovation and Technology (BMVIT), managed by the FFG (Project number: 850457).





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
