# Peer review of "Photoacoustic hygrometer for icing wind tunnel water content measurement: Design, analysis and intercomparison"

_Atmospheric Measurement Techniques, 2020_

## Referee Comment (RC1) · Anonymous Referee #2 · 1 Oct 2020

**1   Content**

This manuscript describes the instrument design and realization of a photoacoustic water vapor and ice water content instrument for operation in a wind tunnel. Further, die instrument is calibrated to a self build humidity generator and compared to reference instruments in the wind tunnel.

**2   Overall impression and rating**

The overall impression of the manuscript is really good. The analysis is done in a balanced way and all aspects important for an instrument manuscript are considered. The presentation of the manuscript is excellent and nicely to read. It is well organized and the analysis and results are clearly structured and communicated in a very detailed way. In addition, I really like the honest and transparent way of the limitations and error analysis description. I think this manuscript is an excellent contribution to the scientific community. For these reasons, I recommend publication in AMT.

I have only very few comments/questions which should be considered before preparing the final/revised version.

**3   Specific comments/questions:**

- page 3, lines 23-24, "positioned outside the tunnel and connected by 7 m long heated and thermally insulated PTFE tubing, ":
  PTFE is not the best material for water vapor measurements at very low conditions (<50 ppmv). PTFE behaves similar to a sponge and could contaminate your probed air by outgassing of water vapor especially if you would like to measure strong gradients to low mixing ratios <50ppmv. So if you plan to go to lower mixing ratios, I would recommend to stainless steal as tubing material.

- Figure 1 b):
  I have a general question to the setup of the photoacoustic cell. It looks like there is a dead volume left of the first window after the gas inlet between the window itself and the collimation optic which is not flushed with the actual measurement air. The same is true for the right window. How strong does such dead volumes

influence your water vapor measurement, if there is a strong difference in humidity between actual probed air (low mixing ratio) and the air within the dead volumes (high mixing ratio)?

- Figure 6 and Section 3.2:
  You described in detail the hygrometer calibration and estimated the uncertainties. I have a questions about the stability of the calibration and the repeatability. Would you get the same calibration function/coefficients, if you would do the same calibration with same PA conditions just the next day or week?. Maybe you can add a short description/discussion about the long term stability of your calibration.

- Page 23, lines 7-9, "Differences (residuals) in background humidities measured by the PA system with the modified BWV inlet and the reference humidity sensor were used to identify measurements exhibiting considerable background humidity offset drift (cf. Fig. 12(a) and (b)),...":
  Do you have any idea, why you measure such background humidity offset drift with your PA instrument ? Is this due to the instrument or more the setup within the wind tunnel. I think it would be good to include some hints or discussion about the reason of the drifts. I mean, if the drifts are from the PA instrument itself, those drifts could also influence your TWC measurement.

**4  Technical comments/suggestions:**

- Figure 13:
  Is the naming of the boxes FZRA and FZDZ within the figure correct ? I would expert the opposite labeling because you have larger particles (550-650 $\mu$m) within FZRA conditions, which should lead to higher CWC values compared to FZDZ

with smaller particles (100 $\mu$m). Or is the number concentration of both particle types so different ?

---

## Referee Comment (RC2) · Anonymous Referee #1 · 2 Oct 2020

This is a high quality work, definitely worth publishing. so only have one general and a few specific questions, primary related to the photoacoustic system. My major question is about the minimum detectable concentration of the presented instrument. For some reasons it is rather high. In the literature there are papers about water vapour measuring PA systems with sub-ppm MDC. The authors should compare their systems with other ones and explain the reasons of this deficiency. Further here are my small questions: 1. Page 5, line 8: It is written: "it exhibits minimal line shift with pressure, high absorption cross section". These parameters should be quantified. Also these parameters of the selected absorption line should be compared quantitatively with the parameters of those absorption lines of water vapour which are in this wavelength

range too. 2. Page 5, line 9. The authors apply square wave modulation. They should explain why do they prefer it instead of sinusoidal modulation. 3. Page 5, line 10-11. It is written: "just below the lasing threshold". In my opinion a modulation which has the lower level just above the lasing threshold is preferable, e.g. as far as the lifetime of the laser is concerned, because the laser effect is not destroyed and re-built in each modulation cycle. 4. Page 5, line 22: response time. The authors should quantify the response time. 5. Page 6, line 6: "Optimum measurement pressure is primarily defined by the valve position of the pressure controller, due to flow noise generated at the valve". This is a strange sentence (but of course it can be true) because normally other parameters, such as the pressure dependent sensitivity of the PA system should decide the applied measurement pressure. It should be explained why this is not the case here.

---

## Author Comment (AC2) · 15 Jan 2021

The comment was uploaded in the form of a supplement:
https://amt.copernicus.org/preprints/amt-2020-295/amt-2020-295-AC2-supplement.pdf

---

## Author Response (AR1)

**Authors' Response to the Referees' Comments**
**(Revision of Manuscript amt-2020-295)**

Benjamin Lang          Wolfgang Breitfuss          Simon Schweighart

Philipp Breitegger          Hugo Pervier          Andreas Tramposch          Andreas Klug

Wolfgang Hassler          Alexander Bergmann

January 15, 2021

We thank both referees for their critical assessment of our work, their valuable comments and the opportunity to clarify some aspects and enhance the submitted manuscript. We have tried to address all comments and suggested improvements in the following and have revised the original manuscript accordingly.

Please note that in the below response the referees' comments are quoted in blue colored text. In each reply we have indicated additions to or changes in the original manuscript in red, with the corresponding location in the original manuscript given in brackets. These modifications have also been indicated in red in the revised manuscript.

Once again, we thank the editor and referees for their time and the valuable feedback.

Sincerely (on behalf of the authors),

Benjamin Lang

**Response to Comments from Anonymous Referee #1 (AR1)**

**Opening statement of Anonymous Referee #1:**
   This is a high quality work, definitely worth publishing. So only have one general and a few specific questions, primary related to the photoacoustic system.

**Comment AR1.1**  My major question is about the minimum detectable concentration of the presented instrument. For some reasons it is rather high. In the literature there are papers about water vapour measuring PA systems with sub-ppm MDC. The authors should compare their systems with other ones and explain the reasons of this deficiency.

**Reply to AR1.1**   The authors certainly agree with the referee that some literature reported minimum detectable concentrations (MDCs) for photoacoustic hygrometers are in the sub-ppm range and that the MDC of the presented device, in comparison, is rather high. We have compiled a non-exhaustive list of literature reported values and setups in Table AR1. Two reasons for the higher MDC of the presented PA system can be given:

1. The current system is based on intensity modulation of the laser, as opposed to wavelength modulation, which is applied in many of the conventional and quartz-enhanced photoacoustic water vapor sensing applications that can be found (cf. Table AR1). Wavelength modulation is known to decrease photoacoustic background signal generation, often resulting in improved signal-to-noise ratios. Therefore, switching to a wavelength modulation-based excitation is assumed to decrease the MDC and is planned to be implemented for the presented system in the future.

2. MDCs or NNEAs of photoacoustic hygrometers (and photoacoustic gas sensors in general) are typically calculated by assuming a linear relationship between the measured signal and the water vapor concentration, i.e., assuming constant sensitivity $s$ of the instrument ($\mathrm{MDC} = 3\sigma_\mathrm{n}/s$; e.g., Bozóki et al., 2011; Szakáll et al., 2001; Liu et al., 2009a; Wu et al., 2019). However, the sensitivity of photoacoustic instruments measuring water vapor in air has been shown to vary strongly with the water vapor concentration itself (Bijnen et al., 1996; Tátrai et al., 2015; Lang et al., 2020). Compared to the maximum sensitivity, which is achieved at high concentrations of water vapor, the sensitivity may decrease to approximately $20\,\%$ at low concentrations ($\leq 100\,\mathrm{ppm}$). Thus, a linear extrapolation down to the detection limit using the sensitivity determined at high concentrations may underestimate the true limit of detection by a factor of up to five. In the presented evaluation the reduction in sensitivity has been considered by the nonlinear calibration function based on Lang et al. (2020). Disregarding this reduction in sensitivity for the hygrometer of this work and linearly extrapolating the signal at the maximum sensitivity would result in an MDC of $4.3\,\mathrm{ppm}$ instead of $23\,\mathrm{ppm}$ for an integration time of $1\,\mathrm{s}$. MDCs are also often reported for water vapor measured in nitrogen as buffer gas (cf. Table AR1), where at low water vapor concentrations the sensitivity is considerably higher than in air (Kosterev et al., 2006; Lang et al., 2020).

   A combination of the above reasons is assumed to explain the comparably high limit of detection determined in this work. To address this issue, we have modified/added the following paragraph to the manuscript and have appended Table AR1 in a supplement to the manuscript.

   (P. 11, Line 26): More practical averaging times of $1\,\mathrm{s}$ and $10\,\mathrm{s}$ result in $3\,\sigma_\mathrm{A}$ noise equivalent concentrations of $23\,\mathrm{ppm}$ and $7\,\mathrm{ppm}$, respectively. A comparison to literature reported detection limits of photoacoustic hygrometers is given in the supplement to this work. The implementation of a wavelength modulation scheme of the laser diode is expected to result in a reduction of the background signal noise and a further improvement of the achievable LOD.

Table AR1: Non-exhaustive list of literature reported minimum detectable concentrations (MDCs) for photoacoustic water vapor detection, together with the method and conditions used for the MDC estimation. All MDC values are (re-)evaluated for three standard deviations of the noise ($3\sigma_n$) based on the information available in the respective publication. Lines highlighted in gray indicate instruments where negligible underestimation of the true MDC is expected for water vapor detection in air according to Lang et al. (2020).

| Reference | MDC[a] ($3\sigma_n$, ppm) | Linear extrap.[b] | Buffer gas | MDC calculation[c] Concentration (ppm) | Pressure (hPa) | PAS type[d] | Source mod.[e] | Source type[f] | Integration time[g] (s) |
|---|---|---|---|---|---|---|---|---|---|
| Besson et al. (2006) | 0.024 | yes | $N_2$ | 1-50 | - | PAS | WM | DFB | 10 |
| Bijnen et al. (1996) | 0.10 | no | air | - | - | PAS | IM | IC CO | - |
| Szakáll et al. (2006) | 0.18 | yes | $N_2$ | 0.1-100 | 1000 | PAS | WM | DFB | 1.5 |
| Szakáll et al. (2007) | 0.19 | yes | $N_2$ | 1-100 | 1000 | PAS | WM | DFB | 1.5 |
| Szakáll et al. (2004) | 0.20 | yes | $N_2$ | 0.1-250 | 500 | PAS | WM | DFB | 4 |
| Kosterev et al. (2006) | 0.24 | yes | $N_2$ | 44.2 | 80 | QEPAS | WM | DFB | 1 |
| Szakáll et al. (2009) | 0.29 | yes | $N_2$ | 2-103 | - | PAS | IM | DFB | 4 |
| Yi et al. (2012b) | 0.50 | yes | air | 18160 | ambient | QEPAS | WM | DFB | 1 |
| Wang et al. (2019) | 0.50 | yes | - | 1100 | 1000 | QEPAS | WM | DFB | 0.2 |
| Liu et al. (2009a) | 0.71 | yes | air | 4420 | ambient | QEPAS | WM | DFB | 1 |
| Liu et al. (2015) | 0.75 | yes | air | 8000 | ambient | QEPAS | WM | DFB | 1 |
| Liu et al. (2010) | 0.78 | yes | air | 14600 | ambient | QEPAS | WM | DFB | 1 |
| Tátrai et al. (2015) | 1.4 | no | air | 10 | 200 | PAS | WM | DFB | - |
| Mikkonen et al. (2018) | 1.4 | yes | air | 7000 | 1000 | CEPAS | - | BBSCS | 50 |
| Szakáll et al. (2001) | 2.4 | yes | air | 5-230 | 975 | PAS | IM | DFB | 10 |
| Zhang et al. (2019) | 2.7 | yes | - | 500-1480 | ambient | PAS | WM | DFB | - |
| Shi et al. (2012) | 3 | yes | air | 7300 | - | PAS | IM | DFB | 1 |
| Wang et al. (2020a) | 3.3 | yes | - | 1500 | ambient | QEPAS | WM | DFB | - |
| Liu et al. (2017) | 3.8 | yes | $N_2$ | 1430 | ambient | PAS | WM | DFB | 0.01 |
| Ma et al. (2018) | 4 | yes | air | 2960 | - | QEPAS | WM | DFB | 1 |
| Bozóki et al. (1996) | 10 | yes | air | 10000 | - | PAS | IM | EC DL | - |
| Dang et al. (2018) | 17 | yes | air | 13000 | ambient | QEPAS | WM | DFB | - |
| This work | 23 | no | air | - | 800 | PAS | IM | DFB | 1 |
| Liu et al. (2009b) | 28 | yes | air | 6400 | ambient | QEPAS | WM | DFB | 1 |
| Bozóki et al. (1999) | 39 | yes | air | 4-150 | ambient | PAS | IM | EC DL | 0.48 |
| Kachanov et al. (2013) | 46 | yes | air | 11000 | 1010 | PAS | WM | EC QCL | 0.3 |
| Elefante et al. (2019) | 60 | yes | air | 16000 | 270 | QEPAS | WM | DFB | 0.1 |
| Ma et al. (2015) | 70 | yes | air | 2100 | ambient | QEPAS | WM | DFB | 1 |
| Mao et al. (2016) | 72 | yes | $N_2$ | 1500 | ambient | PAS | WM | TEDFL | - |
| Bozóki et al. (2010) | 81 | yes | air | 2800 | ambient | PAS | WM | DFB | 0.1 |
| Wu et al. (2019) | 96 | yes | $N_2$ | 4000 | 270 | QEPAS | WM | DFB | 1 |
| Liu et al. (2018) | 120 | yes | air | 11700 | ambient | PAS | WM | DFB | 0.03 |
| Elefante et al. (2020) | 126 | yes | air | 16000 | 270 | QEPAS | WM | ICL | 0.2 |
| Hippler et al. (2010) | 218 | yes | air | 22500 | 200 | PAS | WM | EC DL | 16 |
| Yi et al. (2012a) | 450 | yes | air | 27200 | ambient | QEPAS | WM | DFB | 1 |
| Weidmann et al. (2004) | 580 | yes | - | - | - | QEPAS | WM | SGDBRL | 1 |
| Rey and Sigrist (2008) | 750 | yes | air | 4400 | - | PAS | IM | LED | 30 |
| Wang et al. (2020b) | 834 | yes | air | 11460 | ambient | QEPAS | WM | DFB | - |

[a] $\text{MDC} = 3x_w/\text{SNR} = 3x_w/(S/\sigma_n) = 3\sigma_n/s$, with signal amplitude $S$ and sensitivity $s$

[b] Linear extrapolation used in the estimation of the MDC

[c] MDC estimation performed with PA signals determined at given measurement concentration (range) and pressure

[d] Type of PA cell: Conventional resonator (PAS), quartz-enhanced (QEPAS), cantilever-enhanced (CEPAS)

[e] Radiation source modulation: Intensity modulation (IM), wavelength modulation (WM)

[f] Distributed feedback laser diode (DFB), intracavity CO-laser (IC CO), broadband supercontinuum source (BBSCS), external cavity diode laser (EC DL), external cavity quantum cascade laser (EC QCL), tunable erbium-doped fiber laser (TEDFL), interband cascade laser (ICL), sampled grating distributed Bragg reflector laser (SGDBRL), light emitting diode (LED)

[g] For wavelength modulation, single line scan times may be considerably longer than the stated integration time

**Comment AR1.2** Page 5, line 8: It is written: "it exhibits minimal line shift with pressure, high absorption cross section". These parameters should be quantified. Also these parameters of the selected absorption line should be compared quantitatively with the parameters of those absorption lines of water vapour which are in this wavelength range too.

**Reply to AR1.2** To quantify the stated parameters of the chosen line we have added the value of the spectral line intensity in the manuscript. As the air pressure line shift coefficient is only of secondary importance to the presented application with a PA cell operated at constant pressure, we have removed the mentioning of this parameter. To put the line intensity into context and to allow a comparison to other transitions in the $1.38\,\mu m$ absorption band, we have now noted in the manuscript that the selected line exhibits the highest spectral line intensity in this band and have compared the absorption cross section to other lines targeted in photoacoustic water vapor sensing systems.

(P. 5, Line 6): The diode is temperature-controlled to the peak of a ro-vibrational water vapor transition at $1364.68\,nm$ ($7327.68\,cm^{-1}$; $296\,K$), which was chosen based on HITRAN simulations (Gordon et al., 2017) as it exhibits the highest spectral line intensity in the $1.38\,\mu m$ absorption band ($1.86 \times 10^{-20}\,cm\,molec^{-1}$), as well as low interference from other anticipated atmospheric constituents. At the PA cell operating conditions ($308\,K$, $800\,hPa$) and low water vapor concentrations, the selected line has a maximum absorption cross section of $8.01 \times 10^{-20}\,cm^2\,molec^{-1}$. This is similar to and higher than the cross sections around $1368.6\,nm$ and $1392.5\,nm$, respectively ($8.09 \times 10^{-20}\,cm^2\,molec^{-1}$ and $5.99 \times 10^{-20}\,cm^2\,molec^{-1}$), two regions that have been targeted in previous photoacoustic water vapor sensing applications (e.g., Besson et al., 2006; Kosterev et al., 2006; Tátrai et al., 2015).

**Comment AR1.3** Page 5, line 9. The authors apply square wave modulation. They should explain why do they prefer it instead of sinusoidal modulation.

**Reply to AR1.3** For square wave and sinusoidal modulation with equal peak-to-peak laser intensities, the former is theoretically expected to produce a fundamental frequency component that is by a factor of $4/\pi$ (i.e., $27\,\%$) larger (e.g., Szakáll et al., 2009; Saarela et al., 2009). The signal-to-noise ratios for sinusoidal and square wave modulation, however, have not been compared experimentally for the presented setup. To indicate the reason for the preference of the square wave modulation, we have added the following lines to the manuscript.

(P. 5, Line 11): Square wave rather than sinusoidal modulation has been applied, as a higher signal amplitude is theoretically expected for the former (e.g., Szakáll et al., 2009).

**Comment AR1.4** Page 5, line 10-11. It is written: "just below the lasing threshold". In my opinion a modulation which has the lower level just above the lasing threshold is preferable, e.g. as far as the lifetime of the laser is concerned, because the laser effect is not destroyed and re-built in each modulation cycle.

**Reply to AR1.4** The authors appreciate the practical advice given by the referee, which will be considered in future measurements. For the presented application the photoacoustic signal was maximized by setting the lower modulation level to just below the lasing threshold. As we could not find literature references specifically addressing the lifetime or degradation of DFB semiconductor laser diodes when directly modulated to just below the threshold, we have noted in the manuscript that modulation to above the lasing threshold may be advantageous for practical reasons, referencing Bozóki et al. (2011).

(P. 5, Line 11): Modulation of the laser current to just below the threshold current resulted in a maximized photoacoustic signal amplitude. It should be noted that modulation to slightly above the threshold current may be advantageous for practical reasons (Bozóki et al., 2011).

[Figure]

Figure AR1: **(a)** Background corrected photoacoustic amplitude and phase during a sequence of recovery and response time measurements, performed by alternately sampling humidified and ambient air with water vapor mole fractions of $18,750\,\mathrm{ppm}$ and $5,570\,\mathrm{ppm}$, respectively. A lock-in integration time of $1\,\mathrm{s}$ was used. **(b)** First segment of (a) with indicated PA signal levels used for the determination of the $63.2\,\%$ response and recovery times.

**Comment AR1.5**   Page 5, line 22: response time. The authors should quantify the response time.

**Reply to AR1.5**   We have now added the response and recovery times of the hygrometer in Section 3.1. Additionally, we have added a plot of an example response time measurement to Appendix D of the manuscript (cf. Fig. AR1).

(P. 11, Line 28): As the $1\,\mathrm{s}$ averaging time precision — equivalent to $14\,\mathrm{mg\,kg^{-1}}$ mass mixing ratio at STP — is sufficient for IWT water content measurement and results in favorable response time, this lock-in integration time is applied in calibration and water content measurements. With a $1\,\mathrm{s}$ averaging time, response and recovery times ($63.2\,\%$ PA signal change) of $\tau_{63}=1.7(2)\,\mathrm{s}$ and $\tau_{63}=2.2(2)\,\mathrm{s}$, respectively, have been determined by alternately sampling humidified zero air and ambient air. Response and recovery times for $90\,\%$ signal change are about four times the stated values of $\tau_{63}$. An example response time measurement is shown in Fig. D1 in Appendix D of this manuscript.

(P. 27, Line 10): Figure D1 shows a sequence of recovery and response time measurements, performed with the described instrument by alternately sampling humidified and ambient air.

**Comment AR1.6**   Page 6, line 6: "Optimum measurement pressure is primarily defined by the valve position of the pressure controller, due to flow noise generated at the valve". This is a strange sentence (but of course it can be true) because normally other parameters, such as the pressure dependent sensitivity of the PA system should decide the applied measurement pressure. It should be explained why this is not the case here.

**Reply to AR1.6**   The authors apologize for the ambiguous wording. To clarify, we have rephrased the paragraph and have added Fig. AR2 to the appendix (Appendix A in the revised manuscript), which indicates the PA hygrometer sensitivity and determined SNRs as a function of the PA cell pressure. In the appendix we now also elaborate that the main background noise component at low pressures arises from flow noise generated at the pressure controller valve and that this noise increases with decreasing pressure, presumably due to the position of the valve.

(P. 6, Line 5): The sensitivity of the PA hygrometer, however, is maximized towards higher cell pressures (cf. Appendix A). For IWT measurement, the cell pressure is set to $800(8)\,\mathrm{hPa}$, close to the

[Figure]

Figure AR2: PA signal and signal-to-noise ratio as a function of the pressure of the PA cell operated at $35\,°C$. Signal measurements were performed with humidified air at a water vapor mole fraction of $18,760(120)\,\text{ppm}$ and with an averaging time of $1\,\text{s}$. Noise used in the SNR calculation has been determined from background signal measurements.

pressure of optimal signal-to-noise ratio (SNR) at approximately $850\,\text{hPa}$ (cf. Fig. A1). A lower than optimum cell pressure was used during measurements to allow for the occurring head loss at high IWT airspeeds and TW sampling flow rates.

(P. 26, Line 3): The optimum operating point of the hygrometer in terms of cell pressure has been determined from PA signal measurements acquired with humidified air at a constant water vapor mole fraction of $18,760(120)\,\text{ppm}$ (cf. Fig. A1). As the measured signal, to a first approximation, is proportional to the hygrometer sensitivity, maximum sensitivity can be seen to be achieved towards high cell pressures. Decreasing sensitivity towards lower pressures mainly is a result of decreasing photoacoustic conversion efficiency (Lang et al., 2020), but may also be caused by a lowered sensitivity of the microphone or a lowered overlap of the laser optical emission spectrum with the (narrowing) water vapor absorption line (Bozóki et al., 2003). Signal-to-noise ratios calculated from the measured signals and the noise determined during background signal measurement indicate an optimum operating pressure around $850\,\text{hPa}$ (cf. Fig. A1). By comparison of the noise level determined with and without flow during background signal measurement, noise at low pressures could mainly be attributed to flow noise, which increases with decreasing cell pressure, presumably due to the position of the valve of the pressure controller upstream of the cell.

**Response to Comments from Anonymous Referee #2 (AR2)**

**Opening statement of Anonymous Referee #2:**

This manuscript describes the instrument design and realization of a photoacoustic water vapor and ice water content instrument for operation in a wind tunnel. Further, the instrument is calibrated to a self build humidity generator and compared to reference instruments in the wind tunnel.

The overall impression of the manuscript is really good. The analysis is done in a balanced way and all aspects important for an instrument manuscript are considered. The presentation of the manuscript is excellent and nicely to read. It is well organized and the analysis and results are clearly structured and communicated in a very detailed way. In addition, I really like the honest and transparent way of the limitations and error analysis description. I think this manuscript is an excellent contribution to the scientific community. For these reasons, I recommend publication in AMT.

I have only very few comments/questions which should be considered before preparing the final/revised version.

**Specific comments/questions:**

**Comment AR2.1** page 3, lines 23-24, "positioned outside the tunnel and connected by 7 m long heated and thermally insulated PTFE tubing, ": PTFE is not the best material for water vapor measurements at very low conditions (<50 ppmv). PTFE behaves similar to a sponge and could contaminate your probed air by outgassing of water vapor especially if you would like to measure strong gradients to low mixing ratios <50ppmv. So if you plan to go to lower mixing ratios, I would recommend to stainless steal as tubing material.

**Reply to AR2.1**   The authors thank the referee for pointing out a relevant practical limitation of the instrument, which we have now also indicated in the manuscript at the added discussion regarding the response time of the instrument (cf. reply to Comment AR1.5). In the current work, PTFE instead of stainless steel tubing was used for connecting the probe and the measurement/sampling rack for two reasons. First, the instrument has not been intended for permanent deployment in the particular icing wind tunnel mentioned in the manuscript, but for the application in two separate wind tunnels, which made the use of rigid tubing between the probe and rack impractical. Additionally, as mentioned in Section 3.3, lowest anticipated mole fractions for the presented measurements were in the order of $500\,\mathrm{ppm}$ (dew point of $-30\,°\mathrm{C}$), for which delays due to adsorption and desorption are assumed minor (Wiederhold, 1997).

(P. 11, Line 30): It is noted that response and recovery times of the described setup are assumed longer for measurements of background or total water concentrations below $500\,\mathrm{ppm}$ (dew points below $-30\,°\mathrm{C}$) due to adsorption-desorption effects associated with the polar nature of water and the long PTFE tubing connecting the probe and the measurement unit (Wiederhold, 1997).

**Comment AR2.2** Figure 1 b): I have a general question to the setup of the photoacoustic cell. It looks like there is a dead volume left of the first window after the gas inlet between the window itself and the collimation optic which is not flushed with the actual measurement air. The same is true for the right window. How strong does such dead volumes influence your water vapor measurement, if there is a strong difference in humidity between actual probed air (low mixing ratio) and the air within the dead volumes (high mixing ratio)?

**Reply to AR2.2**  The two volumes adjacent to the windows and outside the actual gas path are filled with ambient air and are sealed by gaskets, as well as PTFE thread seal tape. The water vapor concentrations inside the volumes thus are assumed to remain sufficiently constant in between calibrations, with the

result that any acoustic signals generated in the volumes and measured by the microphone inside the actual gas cell remain constant and are subtracted by the phase-correct PA background signal correction. The low background signal measured during zeroing suggests that negligible acoustic signals arrive at the microphone from these background sources. Attenuation of the laser optical power due to absorption also remained constant within the stated bounds of the optical power, which has been verified by monitoring the laser optical power using the photodetector at the position of the thermal powermeter during zeroing. Nevertheless, a slow gas exchange driven by diffusion and a corresponding change in attenuation of the optical power is expected, which is, however, taken into account by calibrating at regular intervals. We have revised the manuscript to include a concise discussion about the addressed issue.

(P. 5, Line 18): The two outermost volumes of the PA cell, on the left hand side enclosed by the optical collimation unit and the Brewster window, are filled with ambient air and are sealed by gaskets, as well as PTFE thread seal tape. Attenuation of the laser optical power due to absorption from water vapor in these volumes remained constant within the above stated bounds of the optical power in between calibrations.

**Comment AR2.3**  Figure 6 and Section 3.2: You described in detail the hygrometer calibration and estimated the uncertainties. I have a questions about the stability of the calibration and the repeatability. Would you get the same calibration function/coefficients, if you would do the same calibration with same PA conditions just the next day or week?. Maybe you can add a short description/discussion about the long term stability of your calibration.

**Reply to AR2.3**  We thank the referee for the possibility to elaborate on the stability and repeatability of the instrument. We have now included Fig. AR3 in the manuscript, in (a) showing the instrument stability during a measurement of $9,620(80)\,\mathrm{ppm}$ water vapor supplied by the instrument's calibration unit over a duration of three hours after calibration (integration time of $1\,\mathrm{s}$). Variations visible in such a stability analysis would be directly reflected in the calibration function/coefficients. More specifically, Fig. AR3(a) shows the relative deviation of the estimated water vapor mole fraction $x_\mathrm{w}$ from a $1\,\mathrm{s}$ running average of the (calculated) mole fraction $\overline{x}_\mathrm{w,HG}^{(1s)}$ supplied by the instrument's humidity generator:

$$\epsilon(x_\mathrm{w}) = \frac{\Delta x_\mathrm{w}}{\overline{x}_\mathrm{w,HG}^{(1s)}} = \frac{x_\mathrm{w} - \overline{x}_\mathrm{w,HG}^{(1s)}}{\overline{x}_\mathrm{w,HG}^{(1s)}} \ . \tag{1}$$

(Using the running average of the calculated mole fraction as an estimate for the actual water vapor concentration in the hygrometer reduces the impact of saturator pressure and temperature sensor noise on the stability analysis of the hygrometer — sensor noise does not result in changes of the concentration supplied to the hygrometer). The estimated concentration remained within $\pm 1.8\,\%$ of the reference concentration and is well within the $\pm 2.4\,\%$ relative uncertainty of the humidity generator ($95\,\%$ coverage) over the time of the measurement. The determined stability is also within the $3.3\,\%$ accuracy determined for the hygrometer. Negative peaks at $0.4\,\mathrm{h}$, $0.8\,\mathrm{h}$ and $1.2\,\mathrm{h}$ are the result of decreased microphone sensitivity due to minor temperature rises of the PA cell, and the observable oscillation with a period of approximately $3\,\mathrm{h}$ correlates strongly with the drift-corrected temperature inside the instrument rack. Stabilizing the temperature inside the rack is, thus, expected to further improve the instrument stability and accuracy.

We have also added a short discussion on the repeatability, which, however, is limited due to a significant drift ($< 2\,\%\,\mathrm{per\,day}$) that is attributed to the non-existent laser power correction of the PA signal. An example repeatability measurement performed on two consecutive days is given in Fig. AR3(b).

(P. 14, Line 9): The short-term stability of the hygrometer during measurement, which is essential to the instrument accuracy in between calibrations, has been evaluated by supplying a steady flow of

[Figure]

Figure AR3: Hygrometer measurement stability. **(a)** Relative deviation of the measured water vapor mole fraction from the reference concentration of $9,620(80)\,\mathrm{ppm}$, supplied by the calibration unit, over time. **(b)** Relative deviation on two consecutive days, measured at a mole fraction of $18,800(160)\,\mathrm{ppm}$ and calculated using the calibration of day one in both measurements. The gray bands mark the relative uncertainty of the water vapor mole fraction provided by the humidity generator ((a) $\pm 2.4\,\%$, (b) $\pm 2.2\,\%$, both $95\,\%$ coverage). The lock-in integration time used for all measurements was $1\,\mathrm{s}$.

humidified air to the PA cell using the instrument calibration unit. The stability measured over a period of three hours is shown in Fig. 8(a), which shows the relative deviation of the estimated water vapor mole fraction from a $1\,\mathrm{s}$ running average of the mole fraction supplied by the instrument's humidity generator over time. Estimated concentrations remained within $\pm 1.8\,\%$ of the reference concentration and are well within the $\pm 2.4\,\%$ relative uncertainty of the humidity generator ($95\,\%$ coverage). The determined stability also is within the $3.3\,\%$ accuracy of the hygrometer. Negative peaks in Fig. 8(a) at $0.4\,\mathrm{h}$, $0.8\,\mathrm{h}$ and $1.2\,\mathrm{h}$ are the result of decreased microphone sensitivity due to minor temperature rises of the PA cell, and the observable oscillation with a period of approximately $3\,\mathrm{h}$ correlates strongly with the drift-corrected temperature inside the instrument rack. Stabilizing the rack temperature is, thus, expected to further improve the instrument stability and accuracy.

The hygrometer is calibrated on a daily basis, as for longer intervals drift in the lower percentage range has been observed in between calibrations. This drift is mainly associated with a drift in the laser power and the non-existent laser power correction of the PA signal (cf. Section 2.1). Therefore, measurement repeatability has been assessed only by an analysis of the stability over intervals of two consecutive days, where no drift greater than $2\,\%$ has been observed (cf. Fig. 8(b)).

**Comment AR2.4** Page 23, lines 7-9, "Differences (residuals) in background humidities measured by the PA system with the modified BWV inlet and the reference humidity sensor were used to identify measurements exhibiting considerable background humidity offset drift (cf. Fig. 12(a) and (b)),...": Do you have any idea, why you measure such background humidity offset drift with your PA instrument ? Is this due to the instrument or more the setup within the wind tunnel. I think it would be good to include some hints or discussion about the reason of the drifts. I mean, if the drifts are from the PA instrument itself, those drifts could also influence your TWC measurement.

**Reply to AR2.4** The authors acknowledge the need for clarification on the reasons behind the observed background humidity offset drift. The causes of the offset have been mentioned in the original manuscript at page 22, lines 4-6: "*The observable offset is mainly attributed to the* [IWT] *humidity sensor accuracy, as well as to gradients in the IWT air temperature and saturation between the measurement locations.*".

Differing variations of the air temperature and, hence, saturation at the two mentioned measurement/sampling locations in the icing wind tunnel are assumed to be the main cause of the dissimilar drifts in the background humidities measured by the BWV inlet of the PA system and by the reference humidity sensor. Varying differences in the air temperature in the order of several tenths of degrees Celsius between the two locations have frequently been observed during measurements. For saturated air around $-5\,°\mathrm{C}$ such temperature differences may result in background water content differences in the order of some $0.1\,\mathrm{g\,m^{-3}}$ and, therefore, explain the observed offset drifts. We have added this information to the discussion of the measurement results.

(P. 23, Line 9): Dissimilar variations in the air temperature and saturation at the two separate background humidity measurement locations are assumed to be the main cause of the observed drift in the offset, as variations in the temperature difference between the two locations in the order of some tenths of degrees Celsius have frequently been encountered during measurement. For saturated air around $-5\,°\mathrm{C}$ these temperature differences may have resulted in background water content differences and observable offset drifts in the order of some tenths of $\mathrm{g\,m^{-3}}$.

**Comment AR2.5** Figure 13: Is the naming of the boxes FZRA and FZDZ within the figure correct ? I would expert the opposite labeling because you have larger particles (550-650 μm) within FZRA conditions, which should lead to higher CWC values compared to FZDZ with smaller particles (100 μm). Or is the number concentration of both particle types so different?

**Reply to AR2.5** The naming in the figure actually is correct. The terms rain and drizzle indeed only characterize the occurring spectrum maximum drop diameter (with significant abundance) and, in connection with aircraft certification specifications, the normalized drop size distributions (Appendix O of EASA CS-25 (2020) and FAA CFR-25 (2019)). To avoid the possibility of confusion, we have now also added the water content ranges present during the probe intercomparison in the description of the measurements:

(P. 20, Line 9): Condensed water contents during the probe intercomparison ranged from approximately $0.5\,\mathrm{g\,m^{-3}}$ to $0.9\,\mathrm{g\,m^{-3}}$ for freezing drizzle, and from $0.2\,\mathrm{g\,m^{-3}}$ to $0.5\,\mathrm{g\,m^{-3}}$ for freezing rain conditions.

**References**

Besson, J.-P., Schilt, S., and Thévenaz, L.: Sub-ppm multi-gas photoacoustic sensor, Spectrochimica Acta Part A: Molecular and Biomolecular Spectroscopy, 63, 899–904, https://doi.org/10.1016/j.saa.2005.10.034, 2006.

Bijnen, F. G. C., Harren, F. J. M., Hackstein, J. H. P., and Reuss, J.: Intracavity CO laser photoacoustic trace gas detection: cyclic $CH_4$, $H_2O$ and $CO_2$ emission by cockroaches and scarab beetles, Applied Optics, 35, 5357, https://doi.org/10.1364/AO.35.005357, 1996.

Bozóki, Z., Sneider, J., Szabó, G., Miklós, A., Serényi, M., Nagy, G., and Fehér, M.: Intracavity photoacoustic gas detection with an external cavity diode laser, Applied Physics B Laser and Optics, 63, 399–401, https://doi.org/10.1007/BF01828745, 1996.

Bozóki, Z., Sneider, J., Gingl, Z., Mohácsi, Á., Szakáll, M., Bor, Z., and Szabó, G.: A high-sensitivity, near-infrared tunable-diode-laser-based photoacoustic water-vapour-detection system for automated operation, Measurement Science and Technology, 10, 999–1003, https://doi.org/10.1088/0957-0233/10/11/304, 1999.

Bozóki, Z., Szakáll, M., Mohácsi, Á., Szabó, G., and Bor, Z.: Diode laser based photoacoustic humidity sensors, Sensors and Actuators, B: Chemical, 91, 219–226, https://doi.org/10.1016/S0925-4005(03)00120-5, 2003.

Bozóki, Z., Szabó, A., Mohácsi, Á., and Szabó, G.: A fully opened photoacoustic resonator based system for fast response gas concentration measurements, Sensors and Actuators, B: Chemical, 147, 206–212, https://doi.org/10.1016/j.snb.2010.02.060, 2010.

Bozóki, Z., Pogány, A., and Szabó, G.: Photoacoustic instruments for practical applications: Present, potentials, and future challenges, Applied Spectroscopy Reviews, 46, 1–37, https://doi.org/10.1080/05704928.2010.520178, 2011.

Dang, H., Ma, Y., Li, Y., and Wan, S.: High-Sensitivity Detection of Water Vapor Concentration: Optimization and Performance, Journal of Russian Laser Research, 39, 95–97, https://doi.org/10.1007/s10946-018-9694-4, 2018.

EASA CS-25: Certification Specifications and Acceptable Means of Compliance for Large Aeroplanes CS-25, Tech. rep., European Aviation Safety Agency, 2020.

Elefante, A., Giglio, M., Sampaolo, A., Menduni, G., Patimisco, P., Passaro, V. M., Wu, H., Rossmadl, H., MacKowiak, V., Cable, A., Tittel, F. K., Dong, L., and Spagnolo, V.: Dual-Gas Quartz-Enhanced Photoacoustic Sensor for Simultaneous Detection of Methane/Nitrous Oxide and Water Vapor, Analytical Chemistry, 91, 12 866–12 873, https://doi.org/10.1021/acs.analchem.9b02709, 2019.

Elefante, A., Menduni, G., Rossmadl, H., Mackowiak, V., Giglio, M., Sampaolo, A., Patimisco, P., Passaro, V. M., and Spagnolo, V.: Environmental monitoring of methane with quartz-enhanced photoacoustic spectroscopy exploiting an electronic hygrometer to compensate the h2o influence on the sensor signal, Sensors (Switzerland), 20, 2935, https://doi.org/10.3390/s20102935, 2020.

FAA CFR-25: US Code of Federal Regulations - Title 14 Part 25, Airworthiness Standards - Transport Category Airplanes, 2019.

Gordon, I. E., Rothman, L. S., Hill, C., Kochanov, R. V., Tan, Y., Bernath, P. F., Birk, M., Boudon, V., Campargue, A., Chance, K. V., Drouin, B. J., Flaud, J. M., Gamache, R. R., Hodges, J. T., Jacquemart, D., Perevalov, V. I., Perrin, A., Shine, K. P., Smith, M. A., Tennyson, J., Toon, G. C., Tran, H., Tyuterev, V. G., Barbe, A., Császár, A. G., Devi, V. M., Furtenbacher, T., Harrison, J. J., Hartmann, J. M., Jolly, A., Johnson, T. J., Karman, T., Kleiner, I., Kyuberis, A. A., Loos, J., Lyulin, O. M., Massie, S. T., Mikhailenko, S. N., Moazzen-Ahmadi, N., Müller, H. S., Naumenko, O. V., Nikitin, A. V., Polyansky, O. L., Rey, M., Rotger, M., Sharpe, S. W., Sung, K., Starikova, E., Tashkun, S. A., Auwera, J. V., Wagner, G., Wilzewski, J., Wcisło, P., Yu, S., and Zak, E. J.: The HITRAN2016 molecular spectroscopic database, Journal of Quantitative Spectroscopy and Radiative Transfer, 203, 3–69, https://doi.org/10.1016/j.jqsrt.2017.06.038, 2017.

Hippler, M., Mohr, C., Keen, K. A., and McNaghten, E. D.: Cavity-enhanced resonant photoacoustic spectroscopy with optical feedback cw diode lasers: A novel technique for ultratrace gas analysis and high-resolution spectroscopy, Journal of Chemical Physics, 133, 44 308, https://doi.org/10.1063/1.3461061, 2010.

Kachanov, A., Koulikov, S., and Tittel, F. K.: Cavity-enhanced optical feedback-assisted photo-acoustic spectroscopy with a 10.4 $\mu$m external cavity quantum cascade laser, Applied Physics B: Lasers and Optics, 110, 47–56, https://doi.org/10.1007/s00340-012-5250-z, 2013.

Kosterev, A. A., Tittel, F. K., Knittel, T. S., Cowie, A., and Tate, J. D.: Trace Humidity Sensor based on Quartz-Enhanced Photoacoustic Spectroscopy, Lacsea, pp. Paper TuA2 1–3, 2006.

Lang, B., Breitegger, P., Brunnhofer, G., Prats Valero, J., Schweighart, S., Klug, A., Hassler, W., and Bergmann, A.: Molecular relaxation effects on vibrational water vapor photoacoustic spectroscopy in air, Applied Physics B: Lasers and Optics, 126, 1–18, https://doi.org/10.1007/s00340-020-7409-3, 2020.

Liu, K., Guo, X., Yi, H., Chen, W., Zhang, W., and Gao, X.: Off-beam quartz-enhanced photoacoustic spectroscopy, Optics Letters, 34, 1594, https://doi.org/10.1364/OL.34.001594, 2009a.

Liu, K., Li, J., Wang, L., Tan, T., Zhang, W., Gao, X., Chen, W., and Tittel, F. K.: Trace gas sensor based on quartz tuning fork enhanced laser photoacoustic spectroscopy, Applied Physics B: Lasers and Optics, 94, 527–533, https://doi.org/10.1007/s00340-008-3233-x, 2009b.

Liu, K., Yi, H., Kosterev, A. A., Chen, W., Dong, L., Wang, L., Tan, T., Zhang, W., Tittel, F. K., and Gao, X.: Trace gas detection based on off-beam quartz enhanced photoacoustic spectroscopy: Optimization and performance evaluation, Review of Scientific Instruments, 81, 103 103, https://doi.org/10.1063/1.3480553, 2010.

Liu, K., Wang, L., Tan, T., Zhang, W., Chen, W., and Gao, X.: Trace-Gas Detection with Off-Beam Quartz Enhanced Photoacoustic Spectroscopy, International Journal of Thermophysics, 36, 1066–1073, https://doi.org/10.1007/s10765-014-1691-4, 2015.

Liu, K., Mei, J., Zhang, W., Chen, W., and Gao, X.: Multi-resonator photoacoustic spectroscopy, Sensors and Actuators, B: Chemical, 251, 632–636, https://doi.org/10.1016/j.snb.2017.05.114, 2017.

Liu, K., Cao, Y., Wang, G., Zhang, W., Chen, W., and Gao, X.: A novel photoacoustic spectroscopy gas sensor using a low cost polyvinylidene fluoride film, Sensors and Actuators, B: Chemical, 277, 571–575, https://doi.org/10.1016/j.snb.2018.09.037, 2018.

Ma, Y., Yu, X., Yu, G., Li, X., Zhang, J., Chen, D., Sun, R., and Tittel, F. K.: Multi-quartz-enhanced photoacoustic spectroscopy, Applied Physics Letters, 107, 021 106, https://doi.org/10.1063/1.4927057, 2015.

Ma, Y. F., Tong, Y., He, Y., Long, J. H., and Yu, X.: Quartz-enhanced photoacoustic spectroscopy sensor with a small-gap quartz tuning fork, Sensors (Switzerland), 18, 2047, https://doi.org/10.3390/s18072047, 2018.

Mao, X., Zhou, X., Gong, Z., and Yu, Q.: An all-optical photoacoustic spectrometer for multi-gas analysis, Sensors and Actuators B: Chemical, 232, 251–256, https://doi.org/10.1016/j.snb.2016.03.114, 2016.

Mikkonen, T., Amiot, C., Aalto, A., Patokoski, K., Genty, G., and Toivonen, J.: Broadband cantilever-enhanced photoacoustic spectroscopy in the mid-IR using supercontinuum, https://doi.org/10.1364/ol.43.005094, 2018.

Rey, J. M. and Sigrist, M. W.: New differential mode excitation photoacoustic scheme for near-infrared water vapour sensing, Sensors and Actuators, B: Chemical, 135, 161–165, https://doi.org/10.1016/j.snb.2008.08.002, 2008.

Saarela, J., Toivonen, J., Manninen, A., Sorvajärvi, T., and Hernberg, R.: Wavelength modulation waveforms in laser photoacoustic spectroscopy, in: Applied Optics, vol. 48, pp. 743–747, OSA - The Optical Society, https://doi.org/10.1364/AO.48.000743, 2009.

Shi, W., Li, G., and Prasad, C.: Compact laser photoacoustic spectroscopy sensor for atmospheric components measurements, vol. 8366, p. 83660N, https://doi.org/10.1117/12.919297, 2012.

Szakáll, M., Bozóki, Z., Kraemer, M., Spelten, N., Moehler, O., and Schurath, U.: Evaluation of a Photoacoustic Detector for Water Vapor Measurements under Simulated Tropospheric/Lower Stratospheric Conditions, Environmental Science & Technology, 35, 4881–4885, https://doi.org/10.1021/es015564x, 2001.

Szakáll, M., Bozóki, Z., Mohácsi, Á., Varga, A., and Szabó, G.: Diode laser based photoacoustic water vapor detection system for atmospheric research, Applied Spectroscopy, 58, 792–798, https://doi.org/10.1366/0003702041389373, 2004.

Szakáll, M., Huszár, H., Bozóki, Z., and Szabó, G.: On the pressure dependent sensitivity of a photoacoustic water vapor detector using active laser modulation control, Infrared Physics & Technology, 48, 192–201, https://doi.org/10.1016/j.infrared.2006.01.002, 2006.

Szakáll, M., Csikós, J., Bozóki, Z., and Szabó, G.: On the temperature dependent characteristics of a photoacoustic water vapor detector for airborne application, Infrared Physics and Technology, 51, 113–121, https://doi.org/10.1016/j.infrared.2007.04.001, 2007.

Szakáll, M., Varga, A., Pogány, A., Bozóki, Z., and Szabó, G.: Novel resonance profiling and tracking method for photoacoustic measurements, Applied Physics B, 94, 691–698, https://doi.org/10.1007/s00340-009-3391-5, 2009.

Tátrai, D., Bozóki, Z., Smit, H., Rolf, C., Spelten, N., Krämer, M., Filges, A., Gerbig, C., Gulyás, G., and Szabó, G.: Dual-channel photoacoustic hygrometer for airborne measurements: background, calibration, laboratory and in-flight intercomparison tests, Atmospheric Measurement Techniques, 8, 33–42, https://doi.org/10.5194/amt-8-33-2015, 2015.

Wang, C., Wang, Z., and Pang, X.: Quartz-Enhanced Photoacoustic Spectroscopy for Four-Component Gas Detection Based on Two Off-Beam Acoustic Microresonators, Frontiers in Physics, 8, 543, https://doi.org/10.3389/fphy.2020.594326, 2020a.

Wang, Z., Chang, J., Wang, C., Feng, Y., Tian, C., Li, H., Feng, Z., Yu, H., Zhang, H., Zhang, X., Tang, L., and Zhang, Q.: Three acoustic microresonator quartz-enhanced photoacoustic spectroscopy for trace gas sensing, Optics Communications, 452, 286–291, https://doi.org/10.1016/j.optcom.2019.07.061, 2019.

Wang, Z., Zhang, Q., Chang, J., Tian, C., Tang, L., Feng, Y., Zhang, H., and Zhang, X.: Quartz-Enhanced Photoacoustic Spectroscopy Based on the Four-Off-Beam Acoustic Micro-Resonator, Journal of Lightwave Technology, 38, 5212–5218, https://doi.org/10.1109/JLT.2020.2998848, 2020b.

Weidmann, D., Kosterev, A. A., Tittel, F. K., Ryan, N., and McDonald, D.: Application of a widely electrically tunable diode laser to chemical gas sensing with quartz-enhanced photoacoustic spectroscopy, Optics Letters, 29, 1837, https://doi.org/10.1364/OL.29.001837, 2004.

Wiederhold, P. R.: Water Vapor Measurement: Methods and Instrumentation, CRC Press, 1997.

Wu, H., Dong, L., Yin, X., Sampaolo, A., Patimisco, P., Ma, W., Zhang, L., Yin, W., Xiao, L., Spagnolo, V., and Jia, S.: Atmospheric CH4 measurement near a landfill using an ICL-based QEPAS sensor with V-T relaxation self-calibration, Sensors and Actuators B: Chemical, 297, 126 753, https://doi.org/10.1016/j.snb.2019.126753, 2019.

Yi, H., Chen, W., Guo, X., Sun, S., Liu, K., Tan, T., Zhang, W., and Gao, X.: An acoustic model for microresonator in on-beam quartz-enhanced photoacoustic spectroscopy, Applied Physics B, 108, 361–367, https://doi.org/10.1007/s00340-012-4988-7, 2012a.

Yi, H., Chen, W., Sun, S., Liu, K., Tan, T., and Gao, X.: T-shape microresonator-based high sensitivity quartz-enhanced photoacoustic spectroscopy sensor, Optics Express, 20, 9187, https://doi.org/10.1364/oe.20.009187, 2012b.

Zhang, H., Tian, C., Wang, Z., and Zhang, X.: Trace double-component gas sensor in photoacoustic spectroscopy based on frequency division multiplexing, Optical and Quantum Electronics, 51, 268, https://doi.org/10.1007/s11082-019-1981-y, 2019.